## REPORT

# Phosphatidylserine regulates plasma membrane repair through tetraspanin-enriched macrodomains

Yang E. Li[1]\*, Dougall M. Norris[1]\*, Fanqian N. Xiao[1], Elvis Pandzic[2], Renee M. Whan[2], Sandra Fok[2], Ming Zhou[3], Guangwei Du[4], Yang Liu[4], Ximing Du[1], and Hongyuan Yang[1,4]

**The integrity of the plasma membrane is critical to cell function and survival. Cells have developed multiple mechanisms to repair damaged plasma membranes. A key process during plasma membrane repair is to limit the size of the damage, which is facilitated by the presence of tetraspanin-enriched rings surrounding damage sites. Here, we identify phosphatidylserine-enriched rings surrounding damaged sites of the plasma membrane, resembling tetraspanin-enriched rings. Importantly, the formation of both the phosphatidylserine- and tetraspanin-enriched rings requires phosphatidylserine and its transfer proteins ORP5 and ORP9. Interestingly, ORP9, but not ORP5, is recruited to the damage sites, suggesting cells acquire phosphatidylserine from multiple sources upon plasma membrane damage. We further demonstrate that ORP9 contributes to efficient plasma membrane repair. Our results thus unveil a role for phosphatidylserine and its transfer proteins in facilitating the formation of tetraspanin-enriched macrodomains and plasma membrane repair.**

## Introduction

Highly enriched in the plasma membrane (PM) of mammalian cells, cholesterol and phosphatidylserine play essential roles in maintaining the structure and function of the PM (Fairn et al., 2011; Maxfield and Tabas, 2005). Both lipids are synthesized in the endoplasmic reticulum (ER) where they are delivered to the PM through vesicular and non-vesicular mechanisms. It is now recognized that the non-vesicular transfer of lipids mediated by lipid transfer proteins may represent a dominant means by which cells regulate the transport and distribution of lipids (Wong et al., 2018). The oxysterol binding protein (OSBP) and its related proteins (ORP, for OSBP-related protein) constitute a major family of cellular lipid transfer proteins (Antonny et al., 2018; Du et al., 2015; Olkkonen and Li, 2013; Suchanek et al., 2007; Yang, 2006). There are 12 OSBP/ORP members in humans and 7 (Osh1-7) in the budding yeast *Saccharomyces cerevisiae* (Beh et al., 2001). These proteins all share a conserved ~400 amino acid OSBP-related domain (ORD) found near the C-terminus of OSBP, which can bind and transfer lipid ligands. Mechanistically, OSBP/ORPs utilize the hydrolysis of phosphatidylinositol (PI) 4-phosphate (PI4P) as an energy source to transfer more common lipids such as sterols and phosphatidylserine against a concentration gradient (Chung et al., 2015; de Saint-Jean et al., 2011; Im et al., 2005; Mesmin et al., 2013; Moser von Filseck et al., 2015). For instance, OSBP delivers cholesterol from the ER to the Golgi and brings PI4P back to the ER where it is hydrolyzed by the ER-resident phosphatase Sac1 (Mesmin et al., 2013). The hydrolysis of PI4P supplies energy for OSBP to transfer cholesterol against a concentration gradient (de Saint-Jean et al., 2011; Mesmin et al., 2013). Similarly, ORP5 and ORP8 localize to ER–PM contact sites to transfer phosphatidylserine from the ER to the PM (Chung et al., 2015; Ghai et al., 2017; Moser von Filseck et al., 2015). Recent studies demonstrated that some ORPs may also use the hydrolysis of phosphatidylinositol 4, 5-bisphosphate ($PI(4,5)P_2$) to drive the transport of common lipids such as cholesterol and phosphatidylserine (Ghai et al., 2017; Wang et al., 2019).

The integrity of the PM is critical to cell function and survival (Ammendolia et al., 2021). Many types of damage such as mechanical stress or pore-forming toxins can disrupt the integrity of the PM. Cells have developed several mechanisms to repair the damaged PM, including exocytosis, endocytosis, patching, and scission, all of which involve membrane trafficking/fusion/fission (Ammendolia et al., 2021). One well-studied repair mechanism is mediated by the ESCRT (endosomal sorting complex required for transport) complex (Jimenez et al., 2014). Upon PM damage induced by laser, detergents, necrosis, or pyroptosis, ESCRT components are recruited to the site of damage and mediate the shedding of damaged membranes. This mechanism can repair small wounds on the PM (<100 nm in diameter). A critical aspect during membrane damage and

[1]School of Biotechnology and Biomolecular Sciences, University of New South Wales, Sydney, Australia; [2]Katerina Gaus Light Microscopy Facility, Mark Wainwright Analytical Center, University of New South Wales, Sydney, Australia; [3]Verna and Marrs McLean Department of Biochemistry and Molecular Pharmacology, Baylor College of Medicine, Houston, TX, USA; [4]Department of Integrative Biology and Pharmacology, University of Texas Health Science Center at Houston, Houston, TX, USA.

\*Y.E. Li and D.M. Norris contributed equally to this paper. Correspondence to Hongyuan Yang: h.rob.yang@unsw.edu.au, hongyuan.yang@uth.tmc.edu.

repair is how to constrain the damaged area. In a 2022 report, Tetraspanin-enriched macrodomains (TEMA) were found to form rigid rings around damage sites and restrict the spread of membrane damage (Huang et al., 2022). This enables other mechanisms to repair the damage quickly. The mechanisms and putative mediators regulating the formation of the TEMA ring at damage sites remain to be determined.

In this study, we identify ORP9 as a new lipid transfer protein that regulates phosphatidylserine on the PM. Importantly, we also identify phosphatidylserine-enriched rings surrounding damage sites that are dependent upon phosphatidylserine transport. Finally, we show that both the synthesis and transfer of phosphatidylserine are required for the formation of TEMA rings at damage sites.

## Results and discussion

### ORP9 regulates PM PI4P

OSBP and ORPs are known to regulate the level of cellular phosphoinositides, particularly PI4P. However, the specific subcellular site(s) of action for each ORP remains to be elucidated. We carried out a screen to examine the effect of overexpressing each ORP on PI4P in the PM of HeLa cells (Fig. S1, A and B). While many ORPs reduced PM PI4P to some extent, ORP2, ORP5, and the long and short isoforms of ORP9 (ORP9L/S) caused the most striking reduction. ORP5 is well-known to localize to ER–PM contact sites to deliver phosphatidylserine from ER to the PM in exchange for PI4P and PI(4,5)P$_2$ (Chung et al., 2015; Du et al., 2020; Ghai et al., 2017). ORP2 was also demonstrated to regulate PI4P and PI(4,5)P$_2$ on the PM (Wang et al., 2019). By contrast, the effect of ORP9 on PM PI4P is surprising because ORP9 has been known to function only at internal organelles, including the Golgi and endolysosomes (Kawasaki et al., 2022; Tan and Finkel, 2022; Venditti et al., 2019). We therefore decided to further characterize ORP9's role at the PM.

We confirmed the effect of ORP9 overexpression on PM PI4P and also demonstrated that the overexpression of ORP9 mutants defective in PI4P (HH/AA) or PS (L393D) transfer was unable to reduce PM PI4P (Fig. 1, A and B). In fact, their expression led to a moderate increase in PM PI4P (Fig. 1, A and B), suggesting a possible dominant-negative effect. We also verified these findings by total internal reflection fluorescence (TIRF) microscopy using the fluorescent PI4P reporter, GFP-P4Mx2 (Fig. 1, C and D), whereby WT ORP9L and ORP9S, but not the mutants, dramatically reduced PM PI4P. We then carefully examined the effect of overexpression of all known phosphatidylserine transporters of the ORP family (ORP5, ORP8, and ORP9-11) on PI4P distribution (using GFP-P4Mx2) by TIRF and again found that ORP5 and ORP9L/S exhibited the greatest depletion of PM PI4P (Fig. 1, E and F).

Next, we examined PM PI4P in ORP9 knockout (KO) cells generated by CRISPR (Fig. 2 A). In ORP9 KO cells, there was a significant increase in PM PI4P (Fig. 2, B and C). The increase of PM PI4P in ORP9 KO cells was also confirmed by TIRF (Fig. 2, D and E). Importantly, ORP9L/S or ORP5 overexpression dramatically reduced the increased PM PI4P in ORP9 KO cells (Fig. 2, D and E). Together, these data unveil a previously unappreciated role of ORP9 in regulating PI4P of the PM.

### ORP9 regulates PM phosphatidylserine

Since the known function of ORP9 is to deliver phosphatidylserine from the ER to the Golgi and endolysosomes in exchange for PI4P, we hypothesized that ORP9 also delivers phosphatidylserine to the PM. When cells were fixed and detected by a purified phosphatidylserine sensor, GST-2xPH (Li et al., 2021; Tsuji et al., 2019), PM phosphatidylserine was significantly reduced in ORP9 or ORP5 KO HeLa cells (Fig. 2, F and G). An additive reduction of PM phosphatidylserine was detected in ORP5 and ORP9 double KO cells (Fig. 2, F and G), suggesting a functional redundancy between ORP5 and ORP9. GFP-KRAS$_{G12V}$, which is known to localize to PM microdomains enriched in phosphatidylserine (Kattan et al., 2019), displayed a strong PM pattern in WT HeLa cells but shifted to internal membranes (marked by mCherry-CAAX) in ORP5 or ORP9 KO cells (Fig. 2, H and I), further indicating a reduction of PM phosphatidylserine in the KO cells. Previous studies have shown that decreased PM phosphatidylserine can lead to an increase in the accessible pool of cholesterol in the PM due to reduced phosphatidylserine-dependent PM to ER cholesterol transport (Li et al., 2021; Trinh et al., 2020, 2022). Indeed, we observed a significant increase in the accessible pool of free cholesterol as detected by Neon-ALOD4 in ORP9-deficient cells (Fig. S2, A and B). The cholesterol-binding specificity of Neon-ALOD4 was confirmed by treatment of U2OS cells with cholesterol supplementation (4 µg/ml MCD-cholesterol) and depletion (1% HPCD) (Fig. S2 C). Collectively, these data suggest that ORP9 delivers phosphatidylserine to the PM at the expense of PM PI4P.

### ORP9 associates with the PM

Given the observed effects on PM PI4P and phosphatidylserine, we investigated whether ORP9 associates with the PM. Overexpressed ORP9L and 9S showed a largely cytoplasmic pattern under standard confocal microscopy (Fig. 1 A). We therefore used CRISPR to tag ORP9L at its genomic locus with mNeon-Green (mNG). By confocal microscopy, whilst weak, the mNG-ORP9 signal was reduced when ORP9 was knocked down (Fig. S2, D and E). TIRF microscopy revealed a stronger PM association of the tagged endogenous ORP9 than an mNG empty vector control (Fig. S2, F and G). Consistently, despite its seemingly cytoplasmic distribution, overexpressed ORP9L and 9S also showed enhanced PM association by TIRF microscopy (Fig. S2, H and I).

### ORP9 is recruited to damaged PM and required for its repair

ORP9 has recently been shown to play a key role in the repair of damaged lysosomes through its phosphatidylserine-transporting function in a process named phosphoinositide-initiated membrane tethering and lipid transport (PITT) pathway (Tan and Finkel, 2022). Our data above unveil a new role for ORP9 in delivering phosphatidylserine to the PM in exchange for PI4P. We therefore assessed whether ORP9 might also regulate PM repair. We performed laser-based PM wounding using a scanning confocal microscope equipped with a two-photon laser as described (Jimenez et al., 2014) (Fig. 3 A). These experiments revealed rapid PM damage after 30 s, as indicated by the appearance of the impermeant nucleic acid dye propidium iodide (PI) in wounded sites/cells (Fig. 3 B). Both ORP9L and 9S rapidly

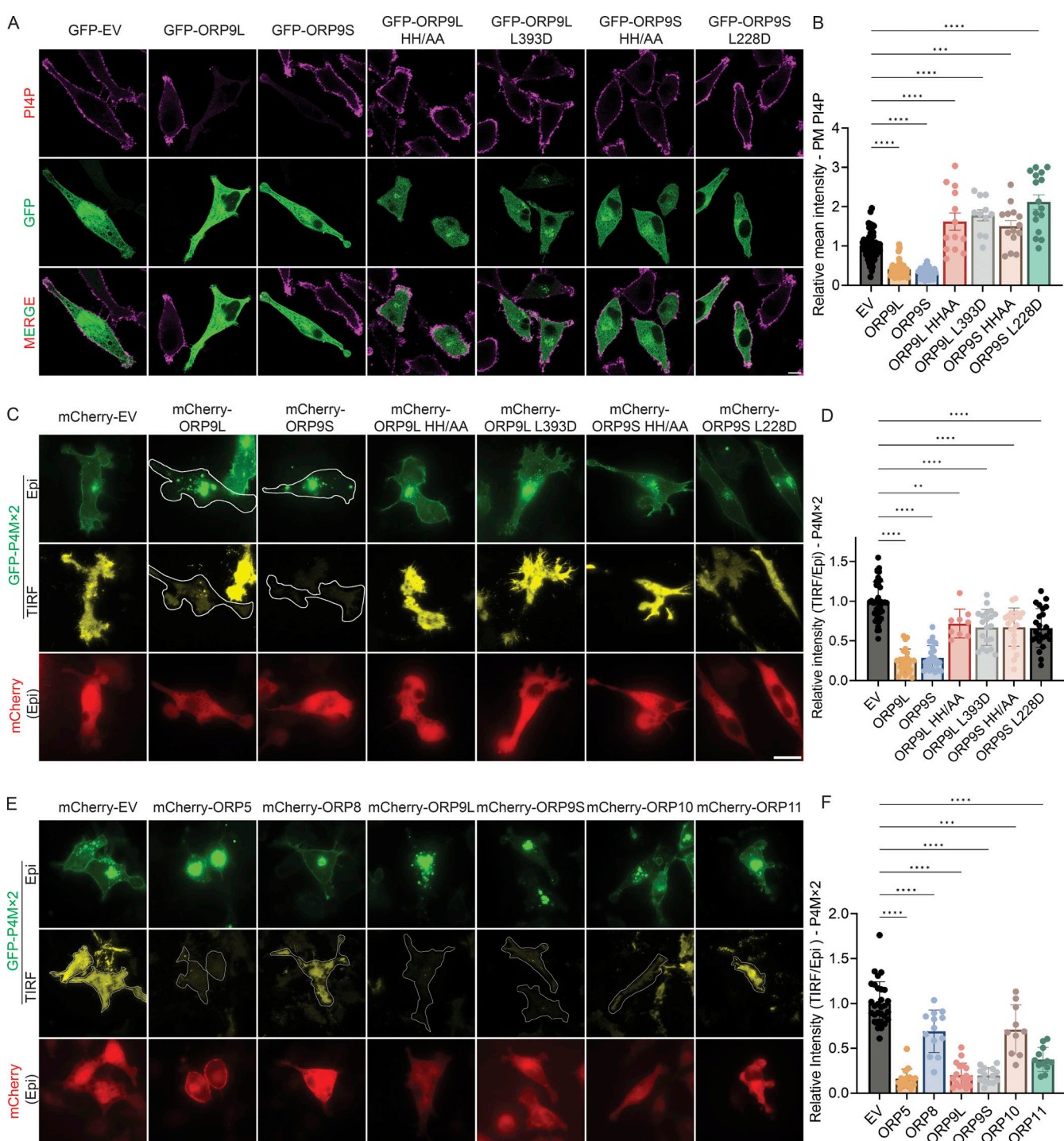

Figure 1.   **Overexpression of ORP9L and ORP9S reduces PM PI4P. (A)** Representative confocal images of PM PI4P in HeLa cells transfected with GFP empty vector (EV), GFP-ORP9L, GFP-ORP9S, GFP-ORP9L HH/AA, GFP-ORP9L L393D, GFP-ORP9S HH/AA, or GFP-ORP9S L228D for 24 h, followed by immunofluorescence with an antibody for PI4P. Scale bars = 10 μm for all images. **(B)** The relative intensity of PM PI4P staining for cells shown in A. ***, P < 0.001; ****, P < 0.0001 (ordinary one-way ANOVA with Dunnett's multiple comparisons test, mean ± SD, $n$ = 11–63 cells per condition). **(C)** Representative TIRF and Epifluorescence (Epi) images of GFP-P4Mx2 in HeLa cells co-transfected with mCherry empty vector (EV), mCherry-ORP9L, mCherry-ORP9S, mCherry-ORP9L HH/AA, mCherry-ORP9L L393D, mCherry-ORP9S HH/AA, or mCherry-ORP9S L228D for 24 h. Scale bars = 20 μm for all images. **(D)** Relative intensity of P4Mx2 (TIRF/Epi) of cells shown in C. **, P < 0.01; ****, P < 0.0001 (ordinary one-way ANOVA with Dunnett's multiple comparisons test, mean ± SD, $n$ = 9–38 cells). **(E)** Representative TIRF and Epi images of GFP-P4Mx2 in HeLa cells co-transfected with mCherry empty vector (EV), mCherry-ORP5, mCherry-ORP8, mCherry-ORP9L, mCherry-ORP9S, mCherry-ORP10, or mCherry-ORP11 for 24 h. Scale bars = 10 μm for all images. **(F)** Relative intensity of P4Mx2 (TIRF/Epi) of cells shown in E. ***, P < 0.001; ****, P < 0.0001 (ordinary one-way ANOVA with Dunnett's multiple comparisons test, mean ± SD, $n$ = 10–30 cells).

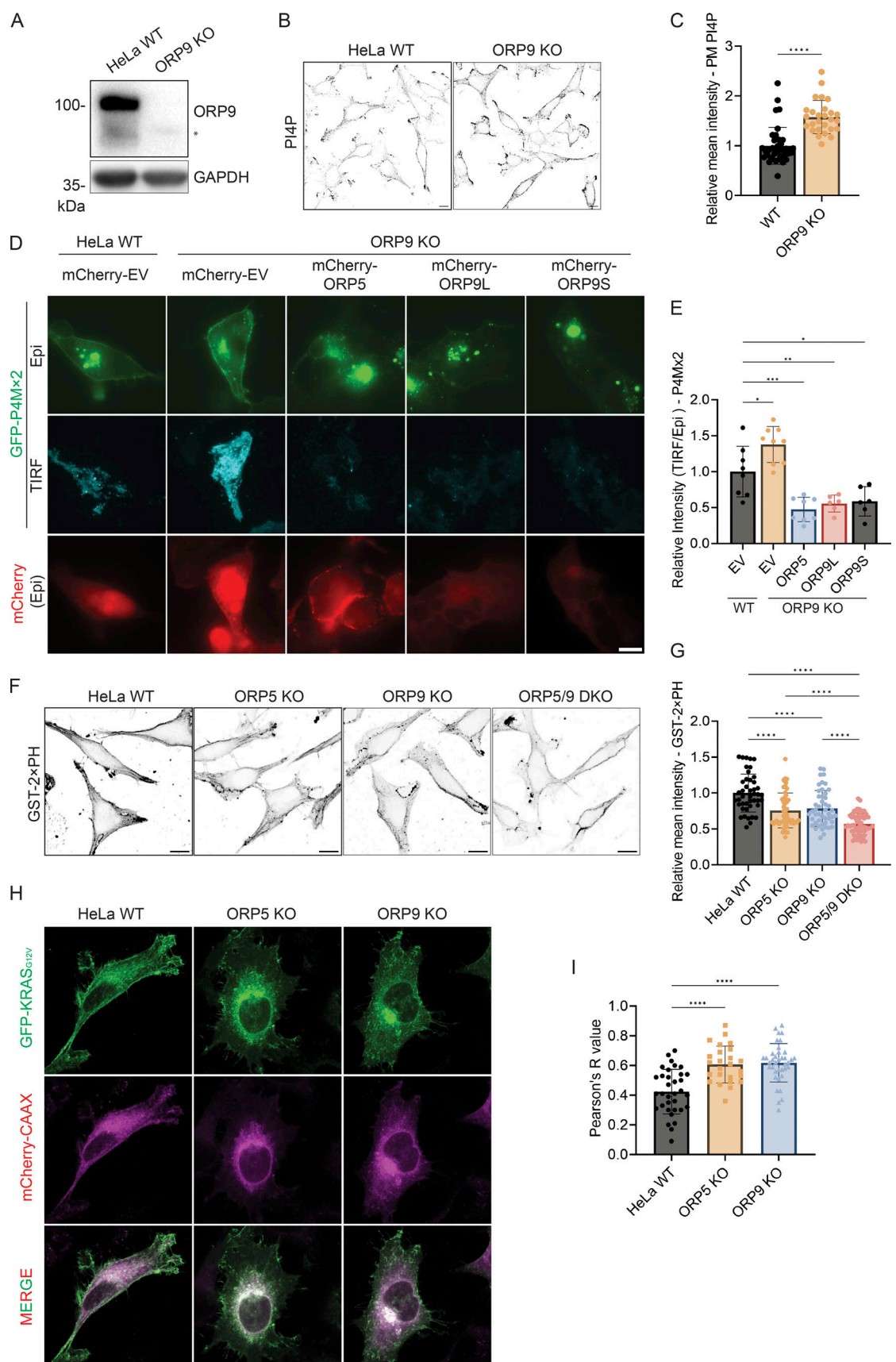

**Figure 2. PM PI4P and PS levels are altered in ORP9-deficient cells. (A)** Validation of the ORP9 KO cell line by Western blot analysis. GAPDH was included as a loading control. *Nonspecific band. **(B)** Representative confocal images of PM PI4P in HeLa WT or ORP9-deficient cells. The PM pool of PI4P was detected

by immunofluorescence with an antibody for PI4P. Scale bars = 10 μm for all images. **(C)** Relative mean intensity of PM PI4P staining for cells shown in B. ****, P < 0.0001 (unpaired *t* test, mean ± SD, n = 27–38 cells). **(D)** Representative TIRF and Epi images of GFP-P4Mx2 in HeLa WT or ORP9-deficient cells co-transfected with mCherry empty vector (EV), mCherry-ORP5, mCherry-ORP9L, or mCherry-ORP9S for 24 h. Scale bars = 10 μm for all images. **(E)** Relative intensity of P4Mx2 (TIRF/Epi) of cells shown in D. *, P < 0.05; **, P < 0.01; ***, P < 0.001 (ordinary one-way ANOVA with Dunnett's multiple comparisons test, mean ± SD, n = 6–9 cells). **(F)** PM PS labeling by GST-2xPH in HeLa WT, ORP5-deficient, ORP9-deficient, or ORP5 and ORP9 double KO cells. **(G)** Relative mean GST-2xPH intensity of cells shown in F. ****, P < 0.0001 (ordinary one-way ANOVA with Dunnett's multiple comparisons test, mean ± SD, n = 44–61 cells). **(H)** Representative confocal images of GFP-KRAS$_{G12V}$ in HeLa WT, ORP5-deficient, or ORP9-deficient cells co-transfected with mCherry-CAAX for 24 h. Scale bars = 10 μm for all images. **(I)** Pearson's R (correlation coefficient) values of GFP-KRAS$_{G12V}$ and mCherry-CAAX for cells shown in H. ****, P < 0.0001 (ordinary one-way ANOVA with Dunnett's multiple comparisons test, mean ± SD, n = 25–38 cells). Source data are available for this figure: SourceData F2.

localized to the damage sites, with ORP9S showing a more striking enrichment (Fig. 3, B and C). By contrast, ORP5, an ER–PM contact site protein with a well-established role in delivering phosphatidylserine to the PM, did not associate with the damage sites (Fig. S3, A and B). ORP9L but not ORP9S displayed a dose-dependent enrichment at damage sites (Fig. S3, C and D), whereby a larger scanning area (maintaining constant laser power) led to a stronger recruitment response. By contrast, ORP5 was lost at damage sites, and the loss of ORP5 at the site of damage was proportional to the size of the damage site in both size and speed (Fig. S3 E). Notably, ORP9 mutants defective in PI4P binding/transfer (HH/AA) or in phosphatidylserine transfer (L/D) showed rather enhanced recruitment to the damage sites (Fig. 3, D–F) (Du et al., 2020), suggesting a possible compensatory response. We further examined the activity of endogenously tagged ORP9L in response to photodamage. Consistently, endogenous mNG-ORP9 was also recruited to the PM at the site of photodamage (Fig. 3, G and H).

In the PITT pathway, ORP9L is recruited to damaged lysosomes by PI4P produced by phosphatidylinositol-4 kinase type 2α (PI4K2A). ORP9L is also known to localize primarily to the Golgi where the level of PI4P is high at steady state. We examined PI4P at PM damage sites using GFP-OSBP-PH, and indeed, there was an enrichment of PI4P (Fig. 3, I and J). Moreover, there was also enrichment of PI4K2A, but not PI4K2B or the GFP control (Fig. 3, K and L). Surprisingly, knocking down PI4K2A had little impact on the recruitment of ORP9L, ORP9S, or OSBP-PH to the PM (Fig. S3, F and G), suggesting there are mechanisms that compensate for the loss of PI4K2A and contribute to the generation of PI4P at damage sites. However, these data suggest that PI4K2A is recruited to sites of PM damage to augment local levels of PI4P, which may be important for membrane repair responses.

A recent study implicated the Golgi apparatus in PM repair (Meng et al., 2023). Since previous work has identified ORP9L on the Golgi apparatus (Venditti et al., 2019), there is a possibility that the ORP9-PM association might be dependent upon the enrichment of Golgi membranes to the site of damage. To examine this, we treated the cells with brefeldin A (BFA), which disrupted the Golgi structure (Fig. S3 H). BFA had little impact on the recruitment of ORP9 (Fig. S3, I and J), which suggests that the recruitment of ORP9 to sites of PM damage occurs independently of a broader Golgi-mediated response.

To examine the role of ORP9 in PM damage repair, we monitored the influx of PI following UV damage. The intensity of cellular PI in ORP9 KO cells rose much faster than that in WT cells during our observation (up to 300 s after injury) (Fig. 4,

A–C). Re-expressing WT ORP9L, and to a lesser extent ORP9S, in ORP9 KO cells impeded the rise in PI intensity (Fig. 4, A–C). Whilst not as effective as WT ORP9L and ORP9S, expression of the phosphatidylserine binding mutants of ORP9L/S conferred a partial resistance of ORP9KO cells to membrane damage compared with ORP9-deficient cells expressing an empty vector control (Fig. 4, A–C). We also demonstrate that ORP9 KO cells have reduced viability upon detergent treatment (Fig. 4 D), suggesting ORP9 can help repair PM damage caused by different stimuli. Next, we wanted to understand mechanistically how ORP9 may regulate the repair of the PM. We first examined the recruitment of ESCRT components, well-known mediators of PM repair, in response to UV damage. CHMP4B recruitment was similar in ORP9 KO and WT cells (Fig. 4, E and F), suggesting ORP9 does not regulate ESCRT recruitment. This is unsurprising, as the recruitment of ESCRT components to damaged lysosomes was not found to be impacted by ORP9/10/11 and OSBP depletion (Tan and Finkel, 2022). Given that the biochemical function of ORP9 is to deliver phosphatidylserine, we first examined the distribution of phosphatidylserine at the sites of UV damage. Strikingly, we observed a strong enrichment of phosphatidylserine/LactC2 at the damage sites, which was disrupted in ORP9 KO cells (Fig. 4, G and H). The enrichment of the phosphatidylserine requires the binding/transfer activity of ORP9 since the phosphatidylserine binding mutants of ORP9L/S failed to fully restore phosphatidylserine recruitment to damage sites in ORP9 KO cells (Fig. 4, I–K). Surprisingly, while not recruited to the damage sites, ORP5 also contributes to the enrichment of phosphatidylserine at the damage sites (Fig. 4, G and H). WT ORP5, but not its phosphatidylserine-binding mutant, restored phosphatidylserine enrichment at PM damage sites (Fig. S3, K and L). Since ORP5 is not enriched at PM damage sites, our data suggest that phosphatidylserine is actively recruited to the damage sites through at least two mechanisms: migration from pre-existing PM pools and delivery from other membranes (the ER).

## Phosphatidylserine facilitates Tspan4 accumulation at damage sites

The phosphatidylserine-enriched ring/patch is reminiscent of a structure formed by Tetraspanin-enriched macrodomains (TEMA) at damage sites, which plays critical roles in limiting the spread of the damage area (Huang et al., 2022). Indeed, the phosphatidylserine-enriched ring and the Tspan4-positive TEMA ring displayed a strong colocalization and appeared to form with similar kinetics (Fig. 5 A). Importantly, the formation of the Tspan4-positive TEMA ring was impaired in ORP9 KO and ORP5 KO cells (Fig. 5, B and C). The TEMA ring was also impaired

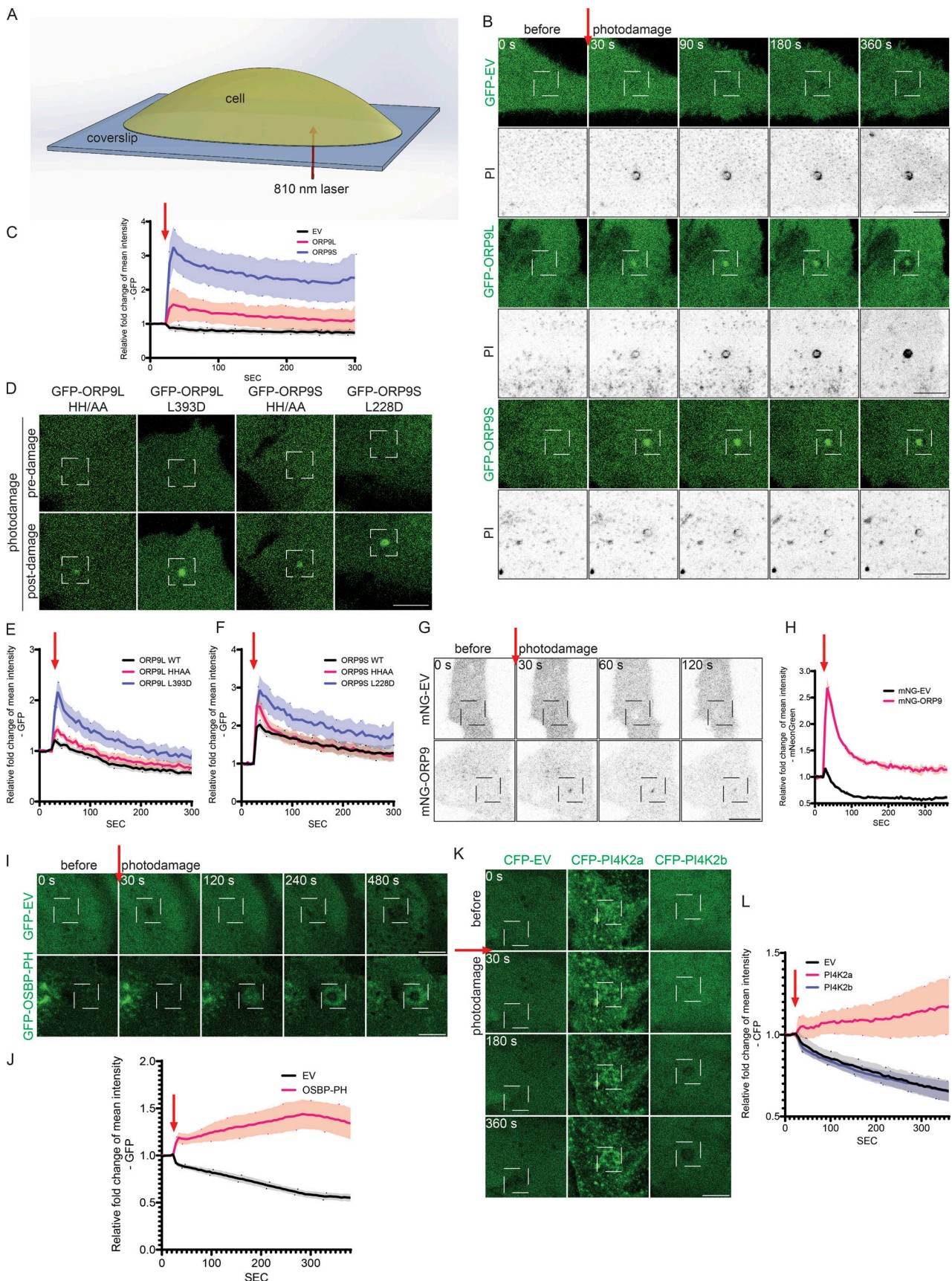

Figure 3. **ORP9 is recruited to damage sites during PM repair. (A)** Schematic of the photodamage assay. **(B)** Representative confocal images of GFP accumulation at the site of UV damage in HeLa cells transfected with GFP empty vector (EV), GFP-ORP9L, or GFP-ORP9S for 24 h. Cells were imaged in phenol

red-free culture medium supplemented with FBS and PI (Propidium Iodide, 50 μg/ml). UV damage occurred 30 s after imaging as indicated by the red arrow with an area of 5 × 5 pixels. Scale bars = 10 μm for all images. **(C)** Relative fold change of GFP accumulation at the site of photodamage as in B. Data shown are the mean ± 95% CI, n = 10–15 cells. **(D)** Representative confocal images of cells transfected with GFP-ORP9L HH/AA, GFP-ORP9L L393D, GFP-ORP9S HH/AA, or GFP-ORP9S L228D for 24 h, Images are shown before and after UV damage, targeting an area of 2 × 2 pixels. Scale bars = 10 μm for all images. **(E and F)** Relative fold change of GFP accumulation at the site of photodamage (2 × 2 pixels) for ORP9L (E) and ORP9S (F) and their mutants as in D. Data shown are the mean ± SEM, n = 7–21 cells. **(G)** Representative confocal images of time-lapse observation of mNeonGreen accumulation in mNeonGreen empty vector (EV) transfected or mNeonGreen-ORP9 KI cells. UV damage occurred 30 s after imaging as indicated by the red arrow with a damage area of 5 × 5 pixels. Scale bars = 10 μm for all images. **(H)** Relative fold change of mNeonGreen accumulation at the damage site (5 × 5 pixels) with photodamage as in G. Data shown are the mean ± SEM, n = 21–35 cells. **(I)** Representative time-lapse confocal images of GFP accumulation in HeLa cells transfected with GFP empty vector (EV) or GFP-OSBP-PH for 24 h. UV damage occurred 30 s after imaging as indicated by the red arrow with a damage area of 5 × 5 pixels. Scale bars = 10 μm for all images. **(J)** Relative fold change of GFP accumulation at the damage site as in I. Data shown are the mean ± 95% CI, n = 21 cells. **(K)** Representative time-lapse confocal images of CFP accumulation in HeLa cells transfected with CFP empty vector (EV) or CFP-PI4K2a or CFP-PI4K2b for 24 h. UV damage occurred 30 s after imaging as indicated by the red arrow with a damage area of 5 × 5 pixels. Scale bars = 10 μm for all images. **(L)** Relative fold change of CFP accumulation at the damage site (5 × 5 pixels) with photodamage as in K. Data shown are the mean ± 95% CI, n = 32–35 cells.

in phosphatidylserine synthase-1 (PSS1) KO cells (Fig. 5, D and E), further indicating a key role for phosphatidylserine in TEMA formation. While our current work focuses on ORP5 and especially ORP9, we hypothesize that other phosphatidylserine transfer proteins, e.g., ORP8, ORP10, and ORP11, may also contribute to PM repair. The role of these proteins will be examined in the future. Together, our findings here suggest that phosphatidylserine can facilitate PM repair by promoting the function of tetraspanins. Phosphatidylserine may be an essential structural component of the TEMA ring. For instance, phosphatidylserine may directly interact with the tetraspanins and promote the assembly and phase separation of TEMA rings at damage sites. Alternatively, phosphatidylserine, especially those with saturated acyl chains, may interact with cholesterol to promote TEMA formation. Cholesterol and Tspan4 are minimal components for the biogenesis of migrasomes, which are evolutionarily conserved extracellular vesicles formed during cell migration (Huang and Yu, 2022; Ma et al., 2015). It will be highly interesting in the future to investigate whether phosphatidylserine regulates migrasome formation/function.

In the PITT pathway, phosphatidylserine facilitates lysosomal repair by recruiting/activating ATG2, a bulk transporter of phospholipids. This then helps generate new membranes and promote the repair of damaged lysosomes. We therefore examined ATG2 recruitment to PM damage sites. ATG2 was indeed also recruited to PM damage sites, but its recruitment appears to be unaffected in ORP9 KO, ORP5 KO, and PSS1 KO cells (Fig. 5, F and G). Thus, mechanistically, the ORP9-mediated PM repair appears to be different from the ORP9-mediated PITT pathway that rapidly repairs damaged lysosomes (Tan and Finkel, 2022). It should be noted that the level of phosphatidylserine at the PM (~20%) is much higher than that of lysosomal membranes (~2%) (Kay and Fairn, 2019). Thus, phosphatidylserine may play different roles in PM and lysosomal repair. Nevertheless, the factors responsible for ATG2 recruitment to PM damage sites and the exact role of ATG2 in PM damage repair require future investigation.

Finally, while our results here implicate a role for phosphatidylserine in the assembly of TEMA rings, direct biochemical evidence for phosphatidylserine-TEMA interaction is lacking. Phosphatidylserine may promote PM repair through mechanisms beyond TEMA formation. Moreover, in ORP5 or ORP9 KO cells, changes in PM lipids other than phosphatidylserine, such

as cholesterol, PI4P, or PI(4,5)P$_2$ could also impair TEMA assembly and PM repair. Future efforts will hopefully elucidate the exact mechanisms by which PM lipids may facilitate PM repair.

## Materials and methods

### Cell culture and transfection
Two strains of Hela cells were involved in this study. HeLa cells (strain M), kindly provided by Dr. Yasunori Saheki from Nanyang Technological University, Singapore, were specifically applied in photodamage experiments given their improved attachment to the coverslip. HeLa cells (strain T) obtained from the American Type Culture Collection were used for the other experiments in this research. U2OS cells were kindly provided by Dr. Edna Hardeman from the University of New South Wales, Australia. Monolayers of cells were cultured in high glucose DMEM, sodium pyruvate (11995065; Thermo Fisher Scientific) supplemented with 10% FBS (Bovogen - SFBS-F) and Pen Strep Glutamine (10378-016; Thermo Fisher Scientific) at 37°C with 5% CO$_2$. Transient plasmid transfection and siRNA transfection were conducted using Lipofectamine LTX/PLUS reagent and Lipofectamine RNAiMAX reagent (Invitrogen) according to the manufacturer's instruction.

### RNAi and cDNA constructs
siRNAs against OSBPL9 (EHU079381) were obtained from Sigma-Aldrich. Details of the cDNA constructs in this study were described previously, including GST-2xPH (GST tagged two of the PH domain of evectin-2 in tandem) (Tsuji et al., 2019), GFP-P4Mx2 (GFP-conjugated tandem fusion of P4M domain consisting 546–647 of *Legionella pneumophila* SidM) (Hammond et al., 2014), Lact-C2-GFP (GFP-conjugated C2 domain of lactadherin) (Yeung et al., 2008), mRFP-Lact-C2 (74061; AddGene), Tspan4-GFP (Huang et al., 2022), CHMP4B-mCherry (116923; AddGene), GFP-KRAS$_{G12V}$, mCherry-CAAX (Kattan et al., 2019), GFP-OSBP, and GFP-ORPs (ORP1 to ORP11) (Wang et al., 2019). GFP/mCherry tagged ORP9L HH/AA, ORP9S HH/AA, ORP9L L393D, and ORP9S L228D were constructed by site-directed mutagenesis.

### Generation of KO/knockin (KI) cell lines
HeLa ORP9 KO, ORP5 KO, ORP5/9 DKO, PSS1 KO, and mNeonGreen-ORP9 KI cells were generated by the CRISPR/Cas9

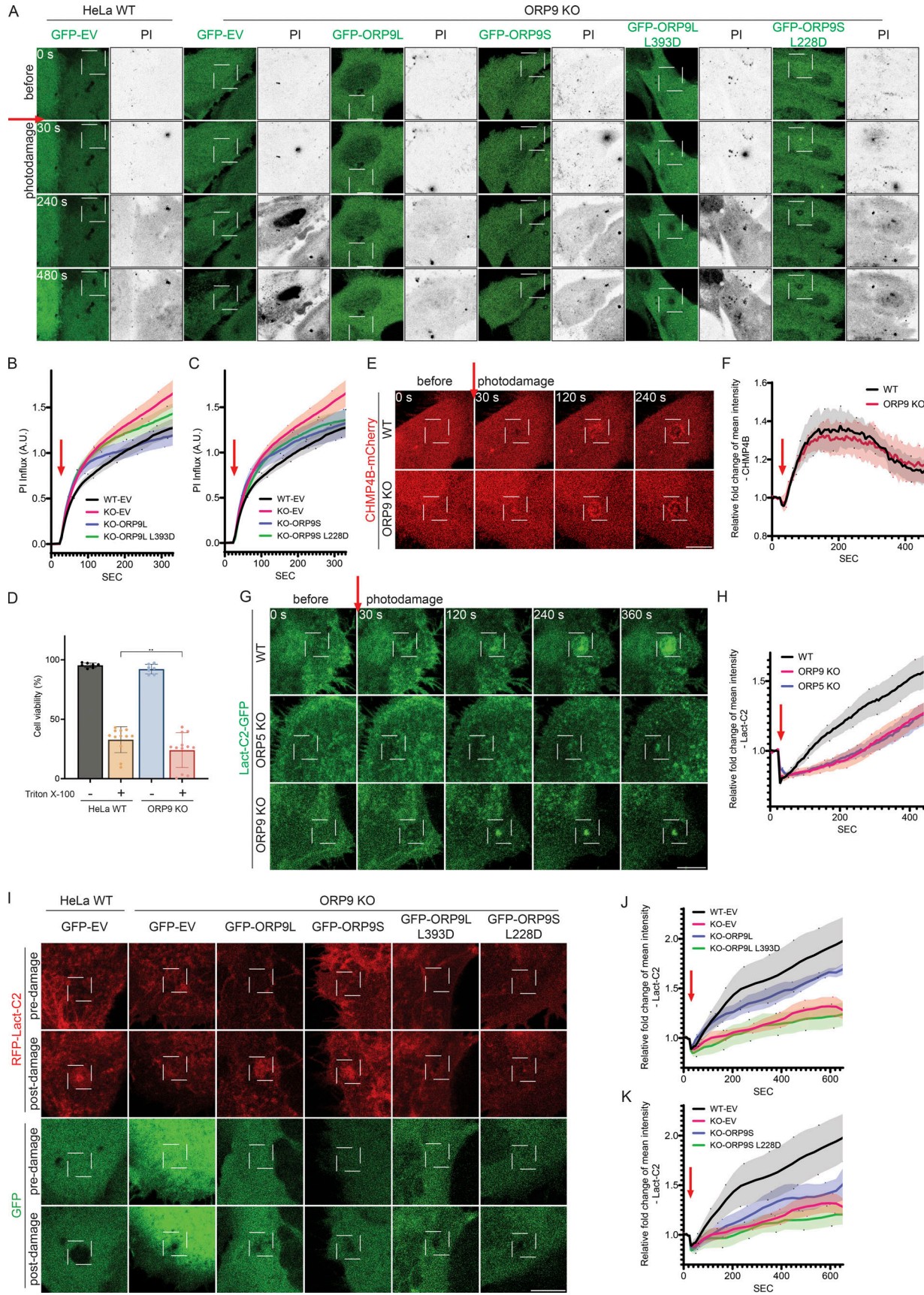

Figure 4. **ORP9 is involved in the regulation of PM repair. (A)** Representative confocal images of GFP accumulation in HeLa WT cells or ORP9-deficient cells transfected with GFP empty vector (EV), GFP-ORP9L, GFP-ORP9S, GFP-ORP9L L393D, or GFP-ORP9S L228D for 24 h. Cells were imaged in phenol

red–free culture medium supplemented with FBS and PI. UV damage occurred 30 s after imaging as indicated by the red arrow with a damage area of 3 × 3 pixels. Scale bars = 10 μm for all images. **(B and C)** Quantification of PI intensity in laser-damaged cells. PI influx for HeLa WT and ORP9-deficient cells expressing GFP-tagged empty vector (EV), ORP9L (B), and ORP9S (C) and their respective mutants as in A, relative to the control (WT-EV). Data shown are the mean ± SEM, n = 37–62 cells. **(D)** Cell viability of HeLa WT or ORP9-deficient cells treated with or without 0.02% Triton X-100 as measured by flow cytometry. \*\*, P < 0.01 (paired t test, mean ± SD, n = 4). **(E)** Representative confocal images of mCherry accumulation in HeLa WT cells or ORP9-deficient cells transfected with CHMP4B-mCherry for 24 h. Cells were imaged in phenol red–free culture medium supplemented with FBS only. UV damage occurred at 30 s with a UV damage size of 5 × 5 pixels. Scale bars = 10 μm for all images. **(F)** Relative fold change of CHMP4B-mCherry accumulation surrounding the photodamage site as in E. Data shown are the mean ± SEM, n = 33–36 cells. **(G)** Representative confocal images of GFP accumulation in HeLa WT cells, ORP5-deficient or ORP9-deficient cells transfected with Lact-C2-GFP for 24 h. UV damage occurred 30 s after imaging as indicated by the red arrow with a damage area of 5 × 5 pixels. Scale bars = 10 μm for all images. **(H)** Relative fold change of Lact-C2-GFP accumulation surrounding the site of photodamage as in G. Data shown are the mean ± SEM, n = 25–32 cells. **(I)** Representative confocal images of RFP and GFP accumulation in HeLa WT cells or ORP9-deficient cells co-transfected with RFP-Lact-C2 and GFP empty vector (EV), GFP-ORP9L, GFP-ORP9L L393D, GFP-ORP9S, or GFP-ORP9S L228D for 24 h. Images are shown for both channels pre and post (10 min) UV damage, targeting an area of 3 × 3 pixels. Scale bars = 10 μm for all images. **(J and K)** Relative fold change of RFP-Lact-C2 accumulation surrounding the site of photodamage, ORP9L (J) and ORP9S (K) and their respective mutants as in I. Data shown are the mean ± SEM, n = 34–48 cells.

system. For KO cell lines, single guide RNAs (sgRNAs) were constructed into pSpCas9(BB)-2A-GFP (PX458) vector, (48138; AddGene). DNA oligonucleotide sequence 5′-CACCGAGCTCGC GGCACGGT-3′ and 5′-TCCGGGGGTCGACTTTCTGA-3′ were chosen as sgRNAs to knock out ORP5. DNA oligonucleotide sequence 5′-AACCATTACTGAAACGTCT-3′ and 5′-CACCAAGAC GTTTCAGTAA-3′ were chosen as sgRNAs to knock out ORP9. DNA oligonucleotide sequence 5′-AGCACTCGGCAAAATTGG GG-3′ and 5′-AAATCTTCGATACGCCACAA-3′ were chosen as sgRNAs to knock out PSS1. HeLa WT cells transfected for 48 h were sorted by BD FACSMelody Cell Sorter (BD Biosciences). A single cell was sorted into each well of 96-well plates. Screening for ORP5 or ORP9 deficiency was determined by Western blot analysis. For mNeonGreen-ORP9 KI cells, sgRNAs were constructed into pSpCas9(BB)-2A-Puro (PX459) vector (62988; AddGene). DNA oligonucleotide sequence 5′-GCTCAGCGGCCC TTCCATGA-3′ were chosen as sgRNAs to knock in mNeonGreen. KI donor plasmid was cloned from the HeLa genome and pCS2-mNeonGreen-C (128144; AddGene). HeLa WT cells transfected for 48 h were treated with puromycin 1 μg/ml for an additional 24 h. The antibiotic-resistant cells were sorted by BD FACSMelody Cell Sorter. The single cell was sorted into each well of 96-well plates. KI efficiency was determined by Western blot analysis.

### Recombinant protein purification

Purification of GST-2xPH was performed as previously described (Li et al., 2021). Briefly, GST-2xPH was expressed in BL21 DE3. Cells were induced with IPTG (0.1 mM), followed by lysozyme lysis and sonication. The recombinant protein was eluted from Pierce Glutathione Superflow Agarose. Purification efficiency was validated by Coomassie blue staining. The concentration of protein was determined by BCA assay.

Purification of Neon-ALOD4 was based on an ALOD4 purification method described earlier (Johnson and Radhakrishnan, 2021). First, BL21(DE3)-competent cells (EC0114; Thermo Fisher Scientific) were transformed with Neon-ALOD4 (kindly provided by the Radhakrishnan Lab, UT Southwestern Medical Center, Dallas, TX, USA), plated on ampicillin agar plates, and incubated overnight at 37°C. Single colonies were picked and then precultured in small scale (12 ml) of Luria-Bertani (LB) broth in a shaking incubator at 37°C until an optical density (OD) at 600 nm of 0.4–0.6 was reached. Cells were then expanded into 100 ml LB broth within an OD of 0.6–0.8, followed by a

further expansion into 1 liter. Once an OD of 0.8 was reached, 1 mM IPTG was added to induce protein production at 18°C shaking at 220 RPM overnight. Cells were then harvested by centrifugation at 4,000 × g at 4°C and resuspended in a homogenization buffer (50 mM Tris-Cl, 150 mM NaCl, 1 mM tris(2-carboxyethyl)phosphine [TCEP], a protease inhibitor cocktail [5892791001; Roche], and 1 mg/ml lysozyme, pH 7.5) and incubated on a rotating mixer for 30 min. The cell suspension was then lysed by sonication on ice for 3 min with a 3-s pulse followed by a 3-s break at 20% amplitude output, followed by a 6-min break, and then repeated two times. The lysate was then clarified at 200,000 × g at 4°C for 1 h. The supernatant was incubated with Ni-NTA Agarose beads (30210; Thermo Fisher Scientific) at 4°C for 1 h. Neon-ALOD4 was eluted from beads with gradient elution buffer (starting with 50 mM imidazole, then 100, 150, 200, and 250 mM imidazole, elution buffer 50 mM Tris-Cl, 150 mM NaCl, and 1 mM TCEP, pH 7.5). The eluate was dialyzed in Slide-A-Lyzer Dialysis Cassettes with a 10,000 kDa cutoff (66380; Thermo Fisher Scientific) in a homogenization buffer at 4°C overnight. The purification efficiency was validated by Coomassie blue staining. The concentration of the protein was determined by the BCA Protein Assay Kit (Sigma-Aldrich).

### Immunofluorescence

Buffers used in this study included washing buffer (PBS containing 50 mM NH₄Cl), buffer A (PBS containing 50 mM NH₄Cl, 5% (vol/vol) NGS, and 0.5% saponin (84510; Sigma-Aldrich), and buffer B (PBS containing 5% NGS and 0.1% saponin).

The PM PI4P/PS labeling was performed following the detailed procedure described earlier (Li et al., 2021). For PM PS immunostaining, cells seeded on glass coverslips (EPBRCSC131GP; Bio-Strategy) were fixed with 4% PFA (C004; Paraformaldehyde, ProSciTech) for 15 min at room temperature (RT). After washing coverslips three times with washing buffer, the procedures were finished on ice. Cells were blocked with buffer A for 45 min, followed by 1 h incubation with GST-2xPH (60 μg/ml), 1 h incubation with rabbit polyclonal to anti-Glutathione-S-Transferase (GST) antibody (G7781, 1:500; Sigma-Aldrich), and 45 min incubation with secondary antibody (1:500; Life Technology). The recombinant protein and the antibodies were diluted in buffer B. Samples were washed three times with PBS between each incubation. Samples were post-fixed with 2% PFA for 10 min on ice and then moved to RT for an additional 5 min, followed by three washes

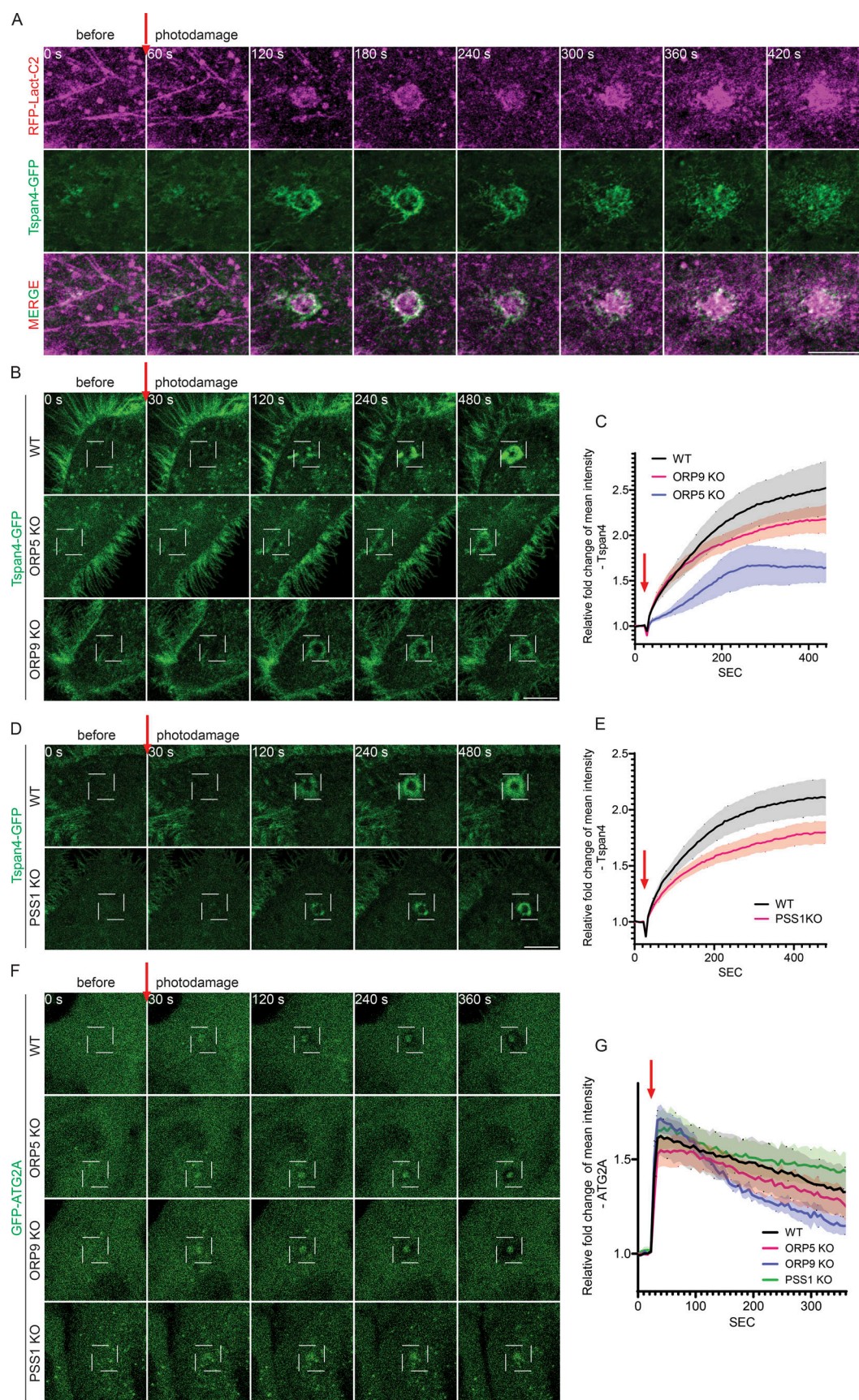

Figure 5. **Phosphatidylserine facilitates Tspan4 accumulation at damage sites. (A)** Representative Airyscan images of time-lapse observation of GFP and RFP accumulation in HeLa WT cells co-transfected with Tspan4-GFP and RFP-Lact-C2 for 24 h. UV damage occurred 60 s after imaging as indicated by the red

arrow with a damage area of 5 × 5 pixels. Scale bars = 10 μm for all images. **(B)** Representative confocal images of GFP accumulation in HeLa WT cells, ORP5-deficient or ORP9-deficient cells transfected with Tspan4-GFP for 24 h. UV damage occurred 30 s after imaging as indicated by the red arrow with a damage area of 5 × 5 pixels. Scale bars = 10 μm for all images. **(C)** Relative fold change of Tspan4-GFP accumulation surrounding the photodamage site as in B. Data shown are the mean ± SEM, *n* = 32–111 cells. **(D)** Representative confocal images of GFP accumulation in HeLa WT cells or PSS1-deficient cells transfected with Tspan4-GFP for 24 h. UV damage occurred 30 s after imaging as indicated by the red arrow with a damage area of 5 × 5 pixels. Scale bars = 10 μm for all images. **(E)** Relative fold change of Tspan4-GFP accumulation surrounding the photodamage site as in D. Data shown are the mean ± SEM, *n* = 44–45 cells. **(F)** Representative confocal images of GFP accumulation in HeLa WT cells, ORP5-deficient, ORP9-deficient, or PSS1-deficient cells transfected with GFP-ATG2A for 24 h. UV damage occurred 30 s after imaging as indicated by the red arrow with a damage area of 5 × 5 pixels. Scale bars = 10 μm for all images. **(G)** Relative fold change of GFP-ATG2A accumulation surrounding the photodamage site as in F. Data shown are the mean ± SEM, *n* = 28–41 cells.

with wash buffer. Coverslips were rinsed with distilled water once before mounting on the slides with Prolong Gold Anti-fade Mounting Media with DAPI (P36935; Thermo Fisher Scientific).

For PM PI4P detection, cells were fixed and blocked as described for the PS labeling protocol above, followed by 1 h incubation with purified PI4P antibody (Z-P004; Echelon) and 45 min incubation with secondary antibody. Cells were post-fixed as indicated.

## PM cholesterol labeling
Live cells were incubated with purified Neon-ALOD4 (3 μM) for 10 min at 37°C and 5% $CO_2$, followed by two PBS washes and 4% PFA fixation for 15 min.

## Brefeldin A treatment
Cells were incubated with Brefeldin A (B7651; 5 μg/ml; Sigma-Aldrich) for 10 min at 37°C with 5% $CO_2$, followed by two PBS washes and 4% PFA fixation for 15 min.

## Microscopy
For TIRF and laser damage experiments cells were seeded on ibidi μ-Slide 8 Well #1.5H Glass Bottom coverslips (Cat no. 80827), and for standard confocal imaging samples, cells were seeded on ibidi μ-Slide 8 Well high coverslips (Cat no. 80806) or mounted coverslips on glass slides. Confocal images were collected by an LSM 900 Airyscan 2 (Zeiss) using Zen Blue acquisition software. TIRF images were collected by Elyra 7 (Zeiss) using Zen Black acquisition software. A 63×/1.4 NA or 40×/1.3 NA oil-immersion objective was applied for imaging at RT. The fluorochromes used for immunofluorescence were Alexa Fluor 488 and Alexa Fluor 568. For comparisons of fluorescence intensities, images were taken during a single session at identical excitation and detection settings.

The photodamage assay was conducted on LSM 880 (Zeiss) using Zen Black acquisition software equipped with a Mai Tai Insight DeepSee multiphoton laser. Cells were maintained in FluoroBrite DMEM (A18967-01; Thermo Fisher Scientific), supplemented with 10% fetal bovine serum (SFBS-F; Bovogen) and Pen Strep Glutamine (10378-016; Thermo Fisher Scientific) at 37°C and 5% $CO_2$, using a Cooling/Heating Insert P Lab-Tek S in a PECON XLmulti S1 Dark LS enclosure equipped with a Zeiss Heating Unit XL S and a $CO_2$ Module S. Images were collected using 63×/1.4 NA oil-immersion objective. The imaging field was magnified x2. The area of interest was damaged using a two-photon laser tuned to 810 nm with 100% output with a pixel dwell time of 84.74 μs for three iterations. The size of the damage region was typically 5 × 5 pixels, or as indicated in the

figure legends. For the mNeonGreen-ORP9 KI cells (Fig. 4, I and J), photodamage was induced using a two-photon laser tuned to 810 nm with 50% output, with a pixel dwell time of 5.3 μs for six iterations across a 5 × 5 pixel region. Time-lapse confocal images were obtained at a continuous frame rate of 6 s.

## Image analysis
For the live-cell photodamage assays investigating the change in protein abundance at the damage site, a circular ROI around the damage site (of a consistent size within each biological replicate) was used to determine mean intensity changes over time. The mean intensity was normalized to the mean intensity of the same region prior to laser damage and therefore expressed as fold-over basal.

For assessment of propidium iodide (PI) influx following laser damage, cell boundaries were defined by thresholding an average projection of the GFP channel over the time course. Following UV damage, the increase in PI intensity (excitation: 561 nm; emission: 569–712 nm) was measured and expressed as the change in intensity relative to the fluorescence prior to laser damage. Individual cellular responses were averaged, providing a mean response for each biological replicate. Biological replicates were normalized to the average intensity across the time-course for the control condition within each replicate, before aggregation, to account for variation in laser power from experiment to experiment.

## Detergent treatment
Cells were harvested and treated with or without 0.02% Triton X-100 for 15 min at room temperature, centrifuged at 500 × *g* for 5 min at 4°C, followed by resuspension in resuspension buffer (0.01 M Hepes/NaOH [pH 7.4], 0.14 M NaCl, 2.5 mM $CaCl_2$) supplied with or without PI (50 μg/ml; Propidium Iodide). Data were analyzed by flow cytometry on BD LSRFortessa Cell Analyzer (BD Biosciences).

## Immunoblot analysis and antibodies
Cells were harvested and resuspended in lysis buffer (1% Triton X-100, 100 mM NaCl, 50 mM Tris, pH 7.8) and incubated for 1 h on ice. Cell lysates were clarified by centrifugation and protein concentration was determined using a BCA protein assay kit. After mixing with 2× Laemmli buffer, samples were subjected to 10% SDS-PAGE for electrophoresis. Proteins were then transferred to a nitrocellulose membrane (#1620115; 0.45 μm; Bio-Rad), followed by blocking with 5% (wt/vol) skim milk in TBST for 1 h at RT. Incubation with primary antibody (1:1,000) was performed overnight at 4°C after three washes for 5 min each in

TBST. Primary antibodies included in this study were GAPDH (2118; Cell Signaling Technology), OSBPL9 (11879-1-AP; Proteintech), and mNeonGreen (55074; Cell Signaling Technology). Secondary antibodies were peroxidase-conjugated AffiniPure donkey anti-rabbit or donkey anti-mouse IgG (H+L; Jackson ImmunoResearch Laboratories) used at a 1:5,000 dilution at RT for 1 h. The bound antibodies were detected by ECL Western blotting detection reagent (GE Healthcare or Merck Millipore) and visualized with Molecular Imager ChemiDocTM XRS+ (Bio-Rad Laboratories).

### Statistical analysis

Statistical analysis was performed by Prism 9 for Windows Ver. 9.0.2 (GraphPad Software) with $t$ test, ordinary one-way ANOVA unless otherwise specified. Data were expressed as mean ± SD, mean ± 95% CI, or mean ± SEM as indicated. For the $t$ tests and one-way ANOVA analyses, data distribution was assumed to be normal, but this was not formally tested.

### Online supplemental material

Fig. S1 shows the screening of OSBP-related proteins (ORPs) in the regulation of PM PI4P. Fig. S2 details the PM cholesterol content in ORP9 KO cells, the tagging of endogenous ORP9 with mNeonGreen, and the PM association of tagged ORP9. Fig. S3 shows the response of ORP5 and ORP9 proteins to laser damage under different perturbations.

### Data availability

The data that support the findings of this study are available within the paper and its supplementary files. Source data used to generate figures containing Western blots have been provided in this study. Reagents and cell lines generated in this work will be shared upon reasonable formal request to the corresponding author.

## Acknowledgments

H. Yang was supported by project grants from the National Health and Medical Research Council (NHMRC) of Australia (1141938, 1141939, and 1144726). H. Yang was a level 3 investigator of the NHMRC (2009852). H. Yang also acknowledges support from the University of New South Wales, Sydney, Australia and the University of Texas Health Science Center at Houston, Houston, TX, USA. Open Access funding provided by University of New South Wales.

Author contributions: Y.E. Li, D.M. Norris, F.N. Xiao, E. Pandzic, S. Fok, and X. Du conducted experiments and analyzed data. R.M. Whan, M. Zhou, Y. Liu, and G. Du analyzed data. Y.E. Li, D.M. Norris, and X. Du also contributed to manuscript writing. H. Yang conceived the project, analyzed data, and wrote the manuscript.

Disclosures: The authors declare no competing interests exist.

Submitted: 12 July 2023

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

# Supplemental material

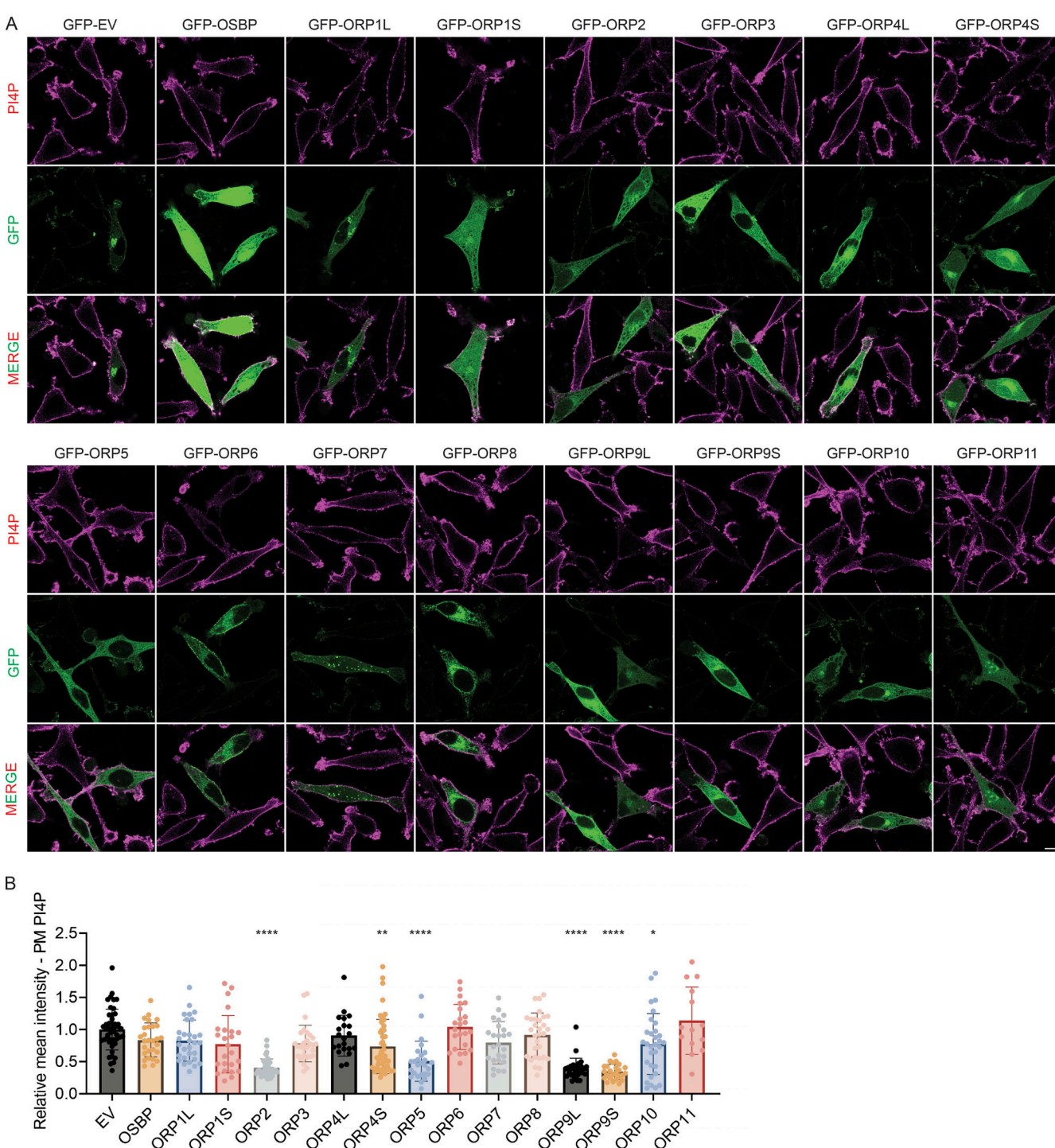

Figure S1. **An overexpression screen of ORP family proteins to assess their role in the regulation of PM PI4P. (A)** Representative confocal images of PM PI4P in HeLa cells transfected with GFP empty vector (EV), GFP-OSBP, GFP-ORP1L, GFP-ORP1S, GFP-ORP2, GFP-ORP3, GFP-ORP4L, GFP-ORP4S, GFP-ORP5, GFP-ORP6, GFP-ORP7, GFP-ORP8, GFP-ORP9L, GFP-ORP9S, GFP-ORP10, or GFP-ORP11 for 24 h, followed by immunofluorescence with an antibody for PI4P. Scale bars = 10 μm for all images. **(B)** Relative intensity of PM PI4P staining for cells shown in A. *, P < 0.05; **, P < 0.01; ****, P < 0.0001 (ordinary one-way ANOVA with Dunnett's multiple comparisons test, mean ± SD, n = 14–48 cells).

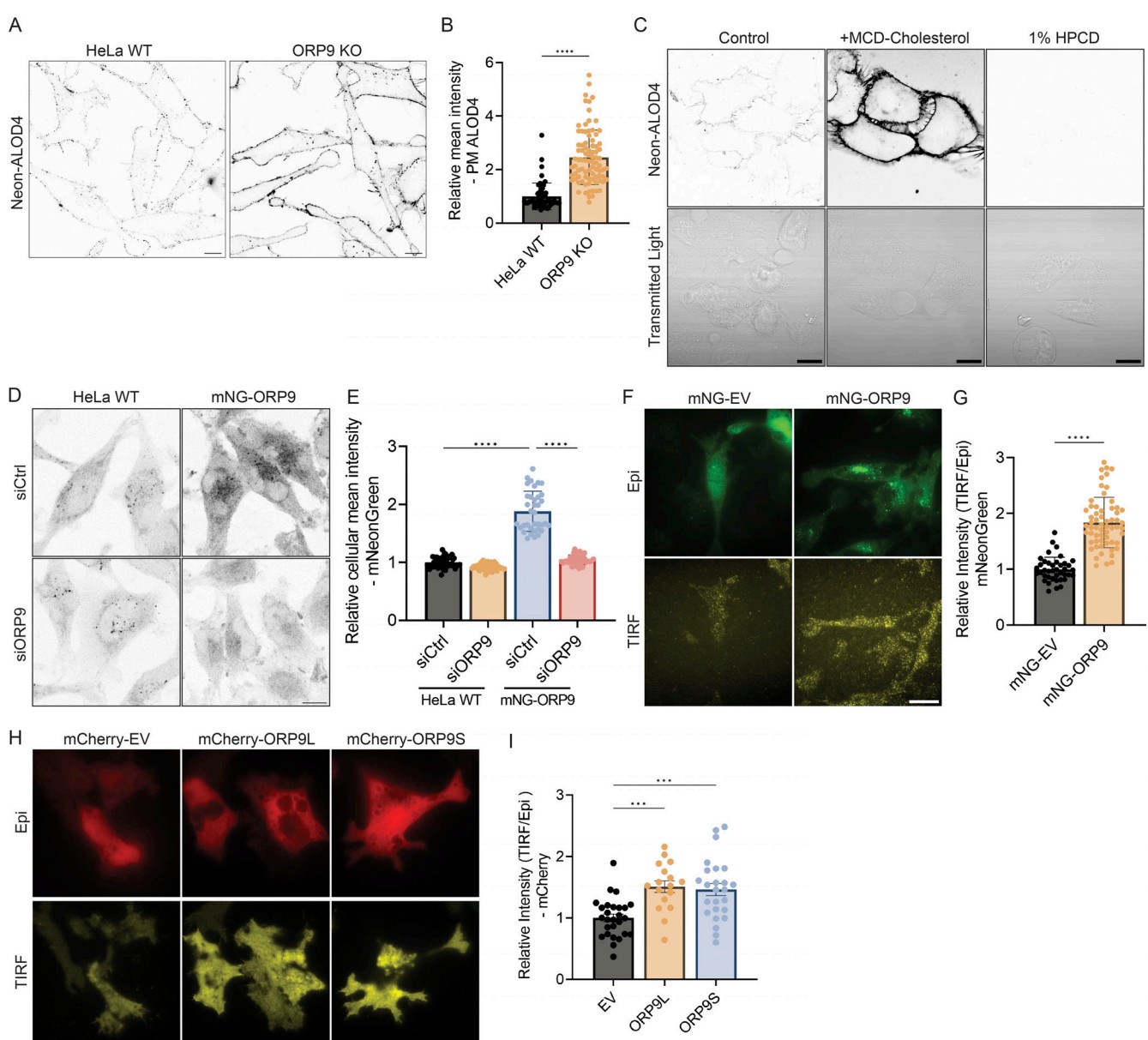

Figure S2. **ORP9 regulates PM cholesterol and partially localizes to the PM. (A)** Representative confocal images of Neon-ALOD4 in HeLa WT or ORP9-deficient cells. Scale bars = 10 µm for all images. **(B)** Relative mean Neon-ALOD4 intensity of cells shown in A. ****, P < 0.0001 (unpaired *t* test, mean ± SD, *n* = 54–86 cells). **(C)** Representative confocal images of Neon-ALOD4 in WT U2OS cells treated with 4 µg/ml MCD-cholesterol supplemented growth medium or DMEM supplemented with 1% HPCD for 1 h. Scale bars = 20 µm for all images. **(D)** Representative confocal images of HeLa WT or mNeonGreen-ORP9 knock-in (KI) cells transfected with control or ORP9 siRNA for 48 h. Scale bars = 10 µm for all images. **(E)** Relative mean cellular intensity of cells shown in D. ****, P < 0.0001 (ordinary one-way ANOVA with Dunnett's multiple comparisons test, mean ± SD, *n* = 36–44 cells). **(F)** Representative TIRF and Epi images of mNeonGreen in mNeonGreen-ORP9 KI cells or HeLa WT cells transfected with mNeonGreen empty vector (EV) for 24 h. Scale bars = 10 µm for all images. **(G)** Relative mean intensity of mNeonGreen (TIRF/Epi) of cells shown in F. ****, P < 0.0001 (unpaired *t* test, mean ± SD, *n* = 40–61 cells). **(H)** Representative TIRF and Epi images of mCherry in HeLa cells co-transfected with mCherry empty vector (EV), mCherry-ORP9L, or mCherry-ORP9S and GFP-P4Mx2 for 24 h. Images were acquired from the same experiment as in Fig. 1 C. Scale bars = 10 µm for all images. **(I)** Relative intensity of mCherry (TIRF/Epi) of cells shown in H. ***, P < 0.001 (ordinary one-way ANOVA with Dunnett's multiple comparisons test, mean ± SD, *n* = 17–27 cells).

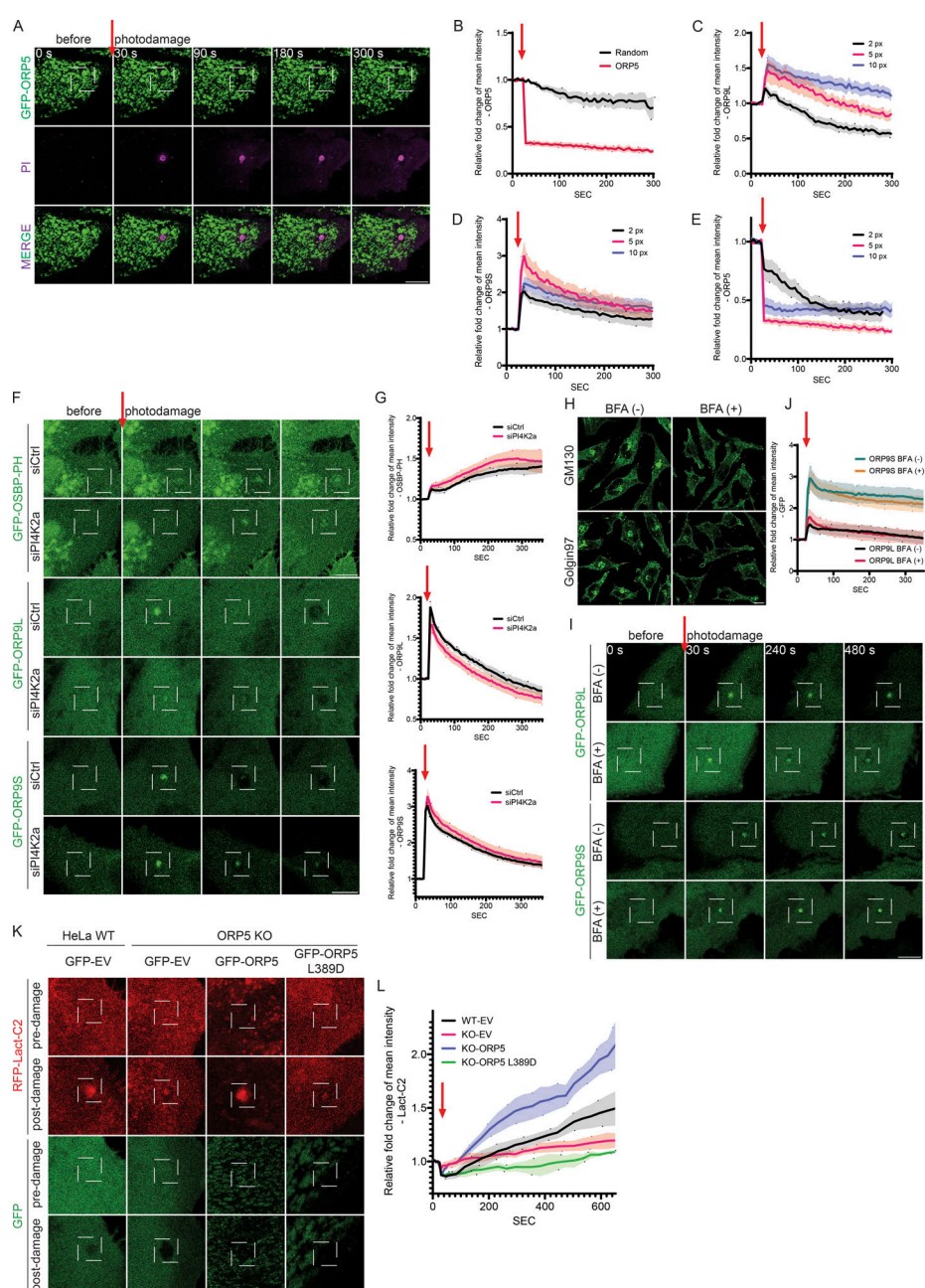

Figure S3. **ORP5 is absent at the site of acute PM damage, and ORP9 accumulation at the damage site is not altered by PI4K2a deficiency or Brefeldin A treatment. (A)** Representative confocal images of GFP accumulation in HeLa cells transfected with GFP-ORP5 for 24 h. Cells were imaged in phenol red–free culture medium supplemented with FBS and PI. UV damage occurred 30 s after imaging as indicated by a red arrow with a damage area of 5 × 5 pixels. Scale bars = 10 μm for all images. **(B)** Relative fold change of GFP accumulation at the photodamage site as in A. Data shown are the mean ± SEM, n = 9 cells. **(C–E)** Relative fold change of GFP-ORP9L (C), GFP-ORP9S (D), and GFP-ORP5 (E) accumulation at the photodamage site in response to an increasing UV damage area (2 × 2 pixels [2 px], 5 × 5 pixels [5 px], and 10 × 10 pixels [10 px]). Data shown are the mean ± SEM, n = 7–21 cells. **(F)** Representative confocal images of GFP accumulation in HeLa cells transfected with siRNA targeting PI4K2a for 24 h and then transfected with GFP-OSBP-PH, GFP-ORP9L, or GFP-ORP9S for 24 h. Cells were imaged in phenol red–free culture medium supplemented with FBS and PI. UV damage occurred 30 s after imaging as indicated by a red arrow with a damage area of 5 × 5 pixels. Scale bars = 10 μm for all images. **(G)** Relative fold change of GFP accumulation at the photodamage site as in F. Data shown are the mean ± SEM, n = 41–52 cells. **(H)** Representative confocal images of Golgi morphology in HeLa WT cells treated with BFA (5 μg/ml) for 10 min, as detected by immunofluorescence using antibodies against GM130 or Golgin97. **(I)** Representative confocal images of GFP accumulation in HeLa cells transfected with GFP-ORP9L or GFP-ORP9S for 24 h, treated with BFA (5 μg/ml) for 10 min prior to the induction of laser damage. Cells were imaged in phenol red–free culture medium supplemented with FBS and PI. UV damage occurred 30 s after imaging as indicated by a red arrow with a damage area of 5 × 5 pixels. Scale bars = 10 μm for all images. **(J)** Relative fold change of GFP accumulation at the photodamage site as in I. Data shown are the mean ± 95% CI, n = 16–18 cells. **(K)** Representative confocal images of RFP and GFP accumulation in HeLa WT cells or ORP5-deficient cells co-transfected with RFP-Lact-C2 and GFP empty vector (EV), GFP-ORP5, or GFP-ORP5 L389D for 24 h. Images are shown for both channels pre and post (10 min) UV damage, targeting an area of 3 × 3 pixels. Scale bars = 10 μm for all images. **(L)** Relative fold change of RFP-Lact-C2 accumulation surrounding the site of photodamage as in K. Data shown are the mean ± SEM, n = 19–30 cells.

