## [Peer Review File · The Journal of Cell Biology]

Phosphatidylserine regulates plasma membrane repair through tetraspanin-enriched macrodomains

Yang Li, Dougall Norris, Fanqian Xiao, Elvis Pandzic, Renee Whan, Sandra Fok, Ming Zhou, Guangwei Du, Yang Liu, Ximing Du, and Hongyuan Yang

Corresponding Author(s): Hongyuan Yang, UNSW Sydney

Review Timeline:

Submission Date:	2023-07-12
Editorial Decision:	2023-08-14
Revision Received:	2023-11-30
Editorial Decision:	2024-01-03
Revision Received:	2024-02-11
Editorial Decision:	2024-02-24
Revision Received:	2024-03-06

Monitoring Editor: Li Yu

Scientific Editor: Andrea Marat

Transaction Report:

DOI: <https://doi.org/10.1083/jcb.202307041>

August 14, 2023

Re: JCB manuscript #202307041

Prof. Hongyuan Yang
UNSW Sydney
School of Biotechnology and Biomolecular Sciences, University of New South Wales
Sydney 2052
Australia

Dear Rob,

Thank you for submitting your manuscript entitled "ORP9 regulates plasma membrane repair by controlling phosphatidylserine and Tetraspanins". The manuscript was assessed by expert reviewers, whose comments are appended to this letter. We invite you to submit a revision if you can address the reviewers' key concerns, as outlined here.

As you will see, the reviewers are all enthusiastic about the description of a novel plasma membrane repair mechanism, especially as combined with recent works on lysosomal repair this suggests lipid transfer as a general mechanism for membrane quality control. They have provided constructive feedback which we hope you find useful. Given the potential impact of your study to open up new research avenues, editorially we agree that it is suitable as Report and therefore a detailed mechanistic investigation can be the subject of further follow up studies. However, we do find it important to further investigate the recruitment of ORP9 and to provide additional evidence for its role in plasma membrane repair. Therefore, in revising we find it important to address these main issues. Specifically:

Regarding investigating the means of ORP9 recruitment (Rev 1p1 and Rev 2p1):

- Examine the localization of lipids in particular PI4P (Rev 1p1, Rev 2p1a), this also seems useful to answer Rev 3p4 regarding PI4P distribution during membrane repair. However, it is unnecessary to examine potential kinases (Rev 2p1a).
- Testing calcium seems unnecessary for the current study (Rev 2p1b).
- Please discuss/hypothesize on the differences between ORP9S and ORP9L (Rev 2p1c).
- Test the suggested potential protein binding partners (Rev 2p1d), examining other potential protein binding partners is likely outside of the scope of the current study.

Determine if ORP9 is recruited by and plays a role in other forms of membrane damage (Rev 2p4).

To provide further evidence for the role of ORP9 and PS in repair:

- Examine ORP9 transfer activity (Rev 2p3a)
- Test the alternative hypothesis that ORP9 is generally altering the membrane composition and that PS could be recruited from other parts of the PM, rather than ORP9 having a more active role with PS being locally transferred from internal membranes (Rev 1p2, Rev 3p2).

Points that do not need to be address experimentally:

- Please discuss cholesterol as suggested by Rev 2p5 (instead of performing additional experiments as suggested by Rev 1p3).
- The internal membrane source of PS (Rev 3p1) and if ORP9 has additional roles for example protein recruitment (Rev 3p3) can be discussed.
- Figure 3 should be removed given the comments of Rev 2p2 and general comment of Rev 1.
- Evidence for how PS promotes repair (Rev 2p3c,d) is not required, though this would be welcomed as it could provide further support for a role of PS.

Otherwise, please address all other reviewer comments in your revised manuscript.

GENERAL GUIDELINES:

Text limits: Character count for a Report is < 20,000, not including spaces. Count includes title page, abstract, introduction, the joint Results & Discussion, and acknowledgments. Count does not include materials and methods, figure legends, references, tables, or supplemental legends.

Figures: Reports may have up to 5 main text figures. To avoid delays in production, figures must be prepared according to the policies outlined in our Instructions to Authors, under Data Presentation, <https://jcb.rupress.org/site/misc/ifora.xhtml>. All figures in accepted manuscripts will be screened prior to publication.

IMPORTANT: It is JCB policy that if requested, original data images must be made available. Failure to provide original images upon request will result in unavoidable delays in publication. Please ensure that you have access to all original microscopy and blot data images before submitting your revision.

Supplemental information: There are strict limits on the allowable amount of supplemental data. Reports may have up to 3 supplemental figures. Up to 10 supplemental videos or flash animations are allowed. A summary of all supplemental material should appear at the end of the Materials and methods section.

Please note that JCB now requires authors to submit Source Data used to generate figures containing gels and Western blots with all revised manuscripts. This Source Data consists of fully uncropped and unprocessed images for each gel/blot displayed in the main and supplemental figures. Since your paper includes cropped gel and/or blot images, please be sure to provide one Source Data file for each figure that contains gels and/or blots along with your revised manuscript files. File names for Source Data figures should be alphanumeric without any spaces or special characters (i.e., SourceDataF#, where F# refers to the associated main figure number or SourceDataFS# for those associated with Supplementary figures). The lanes of the gels/blots should be labeled as they are in the associated figure, the place where cropping was applied should be marked (with a box), and molecular weight/size standards should be labeled wherever possible.

The typical timeframe for revisions is three to four months. While most universities and institutes have reopened labs and allowed researchers to begin working at nearly pre-pandemic levels, we at JCB realize that the lingering effects of the COVID-19 pandemic may still be impacting some aspects of your work, including the acquisition of equipment and reagents. Therefore, if you anticipate any difficulties in meeting this aforementioned revision time limit, please contact us and we can work with you to find an appropriate time frame for resubmission. Please note that papers are generally considered through only one revision cycle, so any revised manuscript will likely be either accepted or rejected.

Thank you for this interesting contribution to Journal of Cell Biology. You can contact us at the journal office with any questions, cellbio@rockefeller.edu or call (212) 327-8588.

Sincerely,

Li Yu, PhD
Monitoring Editor

Andrea L. Marat, PhD
Senior Scientific Editor

Journal of Cell Biology

Reviewer #1 (Comments to the Authors (Required)):

The manuscript by Yang and coworkers investigates the role of ORP9 in maintaining the proper sub-cellular distribution of PS. They uncover that PS or at least Last-C2 form rings around damage sites and that when levels of PS are reduced, the tetraspanin ring around the damage is impaired. The experiments are of high quality. However, my opinion is that the manuscript is underdeveloped even as a "Report."

Given what is known about the other Osh/ORP PS transfer proteins, the results of the first three figures are not surprising.

The three major items that should be addressed.

The observation that ORP9 localizes around damaged plasmalemmal foci is exciting, but the mechanism that would control this particular accumulation is unclear. For instance, if it relies on the PH domain of ORP9, what is the specific lipid ligand or protein

partner that accumulates in this location?

Another issue is that the authors are conflating two observations into a mechanism. ORP9-KO cells have an altered plasma membrane lipid composition, which alone may not support the robust TS4 ring assemblies. This is different from an active role for ORP9 in the process. Since the authors have ORP9-KO cells, they could re-express a "knock-sideways" or degron version of the ORP9 to remove ORP9 function just before the damage induction.

The authors discuss how TS4 and cholesterol are minimal components of the migrasome. However, PS is known to influence the cholesterol content of the PM (Brown & Goldstein, others (PMID: 31227693)). Since ORP9 is a PS-PI4P exchanger, the manuscript should experimentally address the specific role of PS and cholesterol in supporting the TS4 assemblies.

Methods.

The anti-PI4P antibody doesn't pick up the Golgi pool of PI4P. This should be addressed in the text.

For generating the mNG-ORP9 knock-in cells, the authors state they use Cas9. Please confirm this is true and that a Cas9 D10A Nickase wasn't used.

Reviewer #2 (Comments to the Authors (Required)):

In this study, Li et al. identified ORP9 as a new regulator of plasma membrane (PM) repair. Among the 12 ORPs in humans, a few of them including ORP9, when overexpressed, were found to substantially reduce PI4P levels at PM. Interestingly, upon PM damage, ORP9 but not ORP5 is specifically recruited to the damage sites. The phosphatidylserine probe Lact-C2-GFP is also recruited after ORP9, suggesting ORP9-mediated PS transfer to the damage sites. Finally, the authors showed that knockout of ORP9 impaired the enrichment of the PS probe and TSPAN4 at the PM damage sites. This is an interesting study revealing an important part of PM repair. It mechanistically resembles the recently reported ORP9-mediated lysosomal membrane repair and is also consistent with the TEMA model of PM repair. These studies together suggest that lipid transfer might be a general mechanism for the quality control of cellular membranes. An appropriately revised version of this study is expected to provide a much-improved molecular understanding of the PM repair process, unifying the TEMA and PITT pathways.

I have a few comments that I believe will improve the quality of this study.

1. Regarding ORP9 recruitment to the PM damage sites:

- a. ORP9 is usually recruited by PI4P binding to its N-terminal PH domain. Is there increased PI4P (GFP-P4Mx2) at the PM damage sites? If yes, the authors might want to take one step further to determine if such an increase is dependent on PI4K3A or PI4K2A?
- b. Is calcium leakage into the pore required for the recruitment of ORP9 (and potentially PI4P)?
- c. Surprisingly, the authors showed that ORP9S which does not carry a PH domain is recruited to damaged PM more strongly than ORP9L. What are the explanations?
- d. The data suggest that ORP9 is recruited indirectly through other proteins, e.g., ORP10/11 which are known to dimerize with ORP9. Are ORP10/11 recruited to the damage sites? Does ORP9 recruitment require ORP10 or ORP11?

2. The localization of ORP9 to the plasma membrane is minimal if any under basal conditions (Fig. 3C-F). The observed subtle increase in the TIRF signal appears to come from higher expression levels. These data do not support the main conclusions of this paper and are suggested to be removed.

3. Regarding ORP9-mediated PM repair:

- a. Does the repair require the PS transfer activity of ORP9? This can be examined by rescuing experiments using the PS transfer-defective mutants.
- b. Based on the data presented, it is unclear how ORP9-mediated PS transfer promotes PM repair. If the authors propose tetraspanins as PS effectors, more insights into PS regulation of tetraspanins would be helpful in establishing the connection. Is the recruitment of other tetraspanins such as CD63 and CD151 regulated by ORP9?
- c. In the discussion, the authors suggested that due to the high basal levels of PS on the plasma membrane, PS should not be a messenger for PM repair. However, they also argued that PS might be regulating TSPAN4 enrichment at the damage sites. The fact that the PS probe is substantially increased at the damage sites in a ORP9-dependent manner suggests that a regulatory function of PS in PM repair requires much higher local concentrations of PS. It seems likely that similar to lysosomal repair in the PITT pathway, ATG2 might also be activated here by increased local PS levels and the high membrane curvature at the damage pore. The authors can quickly assess EGFP-ATG2 recruitment to the damage sites by live cell imaging in wild type and ORP9-KO cells.

4. Is ORP9 recruited to the membrane pore upon PM damage by detergent or pyroptosis? Functionally, does ORP9-KO render cells vulnerable to detergent treatment or pyroptosis induction?

5. Although not a focus in the current version of this study, the authors might want to discuss potential cholesterol transfer to the PM damage sites by OSBP. The specific enrichment of tetraspanins at the damage ring (TEMAs) suggests much higher local cholesterol levels than undamaged PM.

Minor:

6. Fig. 2B/C/E need to be supported by flow cytometry.

7. Fig. 2D needs quantification.

Reviewer #3 (Comments to the Authors (Required)):

Yang E. Li, Dougall Norris and their colleagues reported a study named "ORP9 regulates plasma membrane repair by controlling phosphatidylserine and Tetraspanins ". It is indeed interesting to learn about their findings regarding ORP9 as a new regulator of phosphatidylserine levels on the plasma membrane and its role in efficient plasma membrane repair. The presence of Tetraspanin-enriched rings and the requirement of functional ORP9 for the formation of both phosphatidylserine- and Tetraspanin-enriched rings highlight a novel mechanism for plasma membrane repair. However, several issues need to be addressed before publishing the study in the Journal of Cell Biology. Here are some possible insights:

Major points:

1. Regarding the origin of phosphatidylserine (PS) and its transfer to the plasma membrane in the context of membrane repair, the study suggests that ORP9 may play a role in delivering PS to the damaged site on the plasma membrane. While the exact internal membrane sources of PS during membrane repair are not specified in the information you provided, it is plausible that PS could be sourced from the endolysosomes, ER, or potentially other internal membrane compartments. In the figure 4K and 4L, the disruption of Golgi apparatus structure by BFA almost did not have impact on the ring structure, do you think PS from endolysosomes or ER would participate in the membrane repair process?
2. ORP9 involved PS transferring to the plasma membrane seems happen under normal state, so how about when the plasma membrane was locally damaged? The PS enriched in the ring structure is locally transferred from internal membrane parts at the precise damage site or laterally recruited from the other parts of plasma membrane.
3. As the authors have mentioned that ORP5 is well known to localize to ER-PM contact sites to deliver phosphatidylserine from ER to the PM in exchange for PI4P and PI(4,5)P2. But, ORP5 didn't participate in the membrane damage repair. So I'm wondering that ORP9 has a specific involvement in the repair mechanism, possibly beyond just the transfer of PS, such as recruiting Tspan proteins and lipids?
4. Referring to the potential ability of ORP9 transferring PS to PM at the expense of PI4P, it is suggested to learn more about the PI4P distribution and concentration level under membrane damage repair process, especially at the ring structure.

Editorial comments:

Given the potential impact of your study to open up new research avenues, editorially we agree that it is suitable as Report and therefore a detailed mechanistic investigation can be the subject of further follow up studies.

Response: many thanks for recognizing the significance and novelty of our work.

Regarding investigating the means of ORP9 recruitment (Rev 1p1 and Rev 2p1):

- Examine the localization of lipids in particular PI4P (Rev 1p1, Rev 2p1a), this also seems useful to answer Rev 3p4 regarding PI4P distribution during membrane repair. However, it is unnecessary to examine potential kinases (Rev 2p1a).

Response: we have now examined PI4P, which is indeed enriched at damage sites (new Figure 3I-J). Although not required, we also examined the potential kinases. We observed enrichment of PI4K2a, but not PI4K2b at damage sites (new Figure 3K-L). However, knocking down PI4K2a did not impact PI4P enrichment and the recruitment of ORP9 (Fig. S3F-3G). We suspect other kinases may be involved. This will be investigated further.

- Testing calcium seems unnecessary for the current study (Rev 2p1b).

Response: Thanks. Agreed.

- Please discuss/hypothesize on the differences between ORP9S and ORP9L (Rev 2p1c).

Response: With its PH domain and FFAT motif, ORP9L is more confined to membrane contact sites. ORP9S, without the PH domain, is more mobile and may serve as a “emergency response unit” that is better suited for quick response to repair membrane damages. ORP9S is very similar to ORP2, which has 4 predicated positively charged patches that mediate PM targeting (see Wang et al., 2019, PMID: 30581148; Fig. 6 and 7). The patches enable ORP9S to bind negatively charged lipids, such as PI4P and PS.

- Test the suggested potential protein binding partners (Rev 2p1d), examining other potential protein binding partners is likely outside of the scope of the current study.

Response: thanks. We respectfully disagree with reviewer 2 that ORP9 is recruited indirectly through other proteins. Both ORP9L and ORP9S can be recruited to the PM by interacting with negatively charged membrane lipids. PI4P is enriched at PM damage sites. Nevertheless, we examined the recruitment of ORP10 and ORP11, and their role in ORP9 function as suggested (see our response to reviewer 2 below). Both ORP10 and ORP11 are efficiently recruited to the damage sites, but they are not required for ORP9 recruitment. We are now carrying out additional experiments to understand the roles of ORP10/11 in PM repair. We hope to exclude the data on ORP10 and 11 from the current manuscript as these data do not improve its core

discoveries. We seek your understanding on this.

Determine if ORP9 is recruited by and plays a role in other forms of membrane damage (Rev 2p4).

Response: we have now demonstrated that ORP9 offers resistance to detergent treatment (new Figure 4C).

To provide further evidence for the role of ORP9 and PS in repair:

- Examine ORP9 transfer activity (Rev 2p3a)

Response: we have examined the PS transfer defective mutants for both ORP9L and ORP9S. Although overexpressed, the PS binding/transfer mutants failed to restore the concentration of phosphatidylserine at the PM (Figure 4H, 4I and 4J).

- Test the alternative hypothesis that ORP9 is generally altering the membrane composition and that PS could be recruited from other parts of the PM, rather than ORP9 having a more active role with PS being locally transferred from internal membranes (Rev 1p2, Rev 3p2).

Response: The highly enriched PS at damage sites is more likely from the ER, where PS is synthesized. ORP proteins are not known to transfer lipids from within the same membrane. In fact, the transfer of PS to the PM by ORPs requires the counter transfer of PI4P from PM to the ER, where PI4P is hydrolyzed. Therefore, although we cannot rule out this possibility, it is highly unlikely that ORP9 extracts PS from other parts of the PM to concentrate it at PM damage sites.

We have now also examined TEMA ring formation at damage sites in cells lacking the major PS synthase, PSS1, which localizes to the ER. In PSS1 KO cells, the formation of TEMA ring is significantly reduced (new Figure 5D), further supporting the core conclusion of this study that phosphatidylserine contributes to PM repair in part through TEMA assembly.

Points that do not need to be address experimentally:

- Please discuss cholesterol as suggested by Rev 2p5 (instead of performing additional experiments as suggested by Rev 1p3).

Response: done. See below.

- The internal membrane source of PS (Rev 3p1) and if ORP9 has additional roles for example protein recruitment (Rev 3p3) can be discussed.

Response: done. See below.

- *Figure 3 should be removed given the comments of Rev 2p2 and general comment of Rev 1.*

Response: we respectfully disagree with Rev 2's comment on this because the association of ORP9 with the PM does occur at steady state, and we have control for differences in ORP9 expression, by normalising the TIRF intensity to whole cell fluorescence intensity. We have decided to keep it as supplemental figures and we seek your understanding on this.

- *Evidence for how PS promotes repair (Rev 2p3c,d) is not required, though this would be welcomed as it could provide further support for a role of PS.*

Response: although not required, we have carried out additional experiments on ATG2A (new Figure 5F-5G). ATG2A was indeed efficiently recruited to the damage sites, but its recruitment is extremely fast and independent of ORP5, ORP9 and PSS1. ATG2 may indeed play an important role in PM repair but this will be examined in future studies.

Otherwise, please address all other reviewer comments in your revised manuscript.

Response: yes, please see below.

Reviewer 1.

The manuscript by Yang and coworkers investigates the role of ORP9 in maintaining the proper sub-cellular distribution of PS. They uncover that PS or at least Last-C2 form rings around damage sites and that when levels of PS are reduced, the tetraspanin ring around the damage is impaired. The experiments are of high quality. However, my opinion is that the manuscript is underdeveloped even as a "Report."

Response: We have added a lot more data to make the study more compelling. Also, as pointed out by the editors, "the potential impact of your study to open up new research avenues, editorially we agree that it is suitable as Report and therefore a detailed mechanistic investigation can be the subject of further follow up studies".

Given what is known about the other Osh/ORP PS transfer proteins, the results of the first three figures are not surprising.

Response: we respectfully disagree. ORP9 is only known to function in internal organelles. This is the first study to unveil its localization and function at the plasma membrane.

The three major items that should be addressed.

The observation that ORP9 localizes around damaged plasmalemmal foci is exciting, but the

mechanism that would control this particular accumulation is unclear. For instance, if it relies on the PH domain of ORP9, what is the specific lipid ligand or protein partner that accumulates in this location?

Response: The membrane targeting of ORP9L is known to depend on PI4P and its PH domain (PMID: 36071159). We have now demonstrated the enrichment of PI4P and PI4K IIA at the damage sites (new figure 3I-3L). PI4P is known to recruit ORP9L.

ORP9S does not have a PH domain and is very similar to ORP2, which has 4 predicted positively charged patches that mediate PM targeting (PMID: 30581148, Fig. 6 and 7). Thus, ORP9S can also be recruited to the damage site through PI4P, and other negatively charged lipids such as PS.

Another issue is that the authors are conflating two observations into a mechanism. ORP9-KO cells have an altered plasma membrane lipid composition, which alone may not support the robust TS4 ring assemblies. This is different from an active role for ORP9 in the process. Since the authors have ORP9-KO cells, they could re-express a "knock-sideways" or degron version of the ORP9 to remove ORP9 function just before the damage induction.

Response: Perhaps there is a misunderstanding here. We are also confused by the term "active" role. In any case, our data support a role for ORP9 and especially phosphatidylserine in the assembly of TEMA rings. The novelty of this study is to demonstrate rapid concentration of phosphatidylserine at PM damage sites and to establish a link between phosphatidylserine and TEMA rings.

The authors discuss how TS4 and cholesterol are minimal components of the migrasome. However, PS is known to influence the cholesterol content of the PM (Brown & Goldstein, others (PMID: 31227693). Since ORP9 is a PS-PI4P exchanger, the manuscript should experimentally address the specific role of PS and cholesterol in supporting the TS4 assemblies.

Response: Thanks for this point. Cholesterol is known to impact the assembly of tetraspanins. Our work here focuses on phosphatidylserine, a previously unknown and therefore novel factor for tetraspanin function. It should also be noted that the level of plasma membrane cholesterol increased in ORP9 KO cells as we showed in Figure S2A-2C. Thus, the disruption of TEMA assembly in ORP9 KO cells is unlikely caused by cholesterol deficiency. Our data suggest an independent role for PS in Tspan4 assembly. As suggested by the editors, the role of cholesterol during PM repair will be carefully examined in follow-up studies, although cholesterol should impact PM repair given its known function in TEMA formation.

Reviewer 2

In this study, Li et al. identified ORP9 as a new regulator of plasma membrane (PM) repair. Among the 12 ORPs in humans, a few of them including ORP9, when overexpressed, were found to substantially reduce PI4P levels at PM. Interestingly, upon PM damage, ORP9 but not ORP5 is specifically recruited to the damage sites. The phosphatidylserine probe Lact-C2-GFP is also recruited after ORP9, suggesting ORP9-mediated PS transfer to the damage sites. Finally, the authors showed that knockout of ORP9 impaired the enrichment of the PS probe and TSPAN4 at the PM damage sites. This is an interesting study revealing an important part of PM repair. It mechanistically resembles the recently reported ORP9-mediated lysosomal membrane repair and is also consistent with the TEMA model of PM repair. These studies together suggest that lipid transfer might be a general mechanism for the quality control of cellular membranes. An appropriately revised version of this study is expected to provide a much-improved molecular understanding of the PM repair process, unifying the TEMA and PITT pathways.

Response: many thanks for these comments. Indeed, PS is important for repairing both PM and lysosomal membranes.

I have a few comments that I believe will improve the quality of this study.

1. Regarding ORP9 recruitment to the PM damage sites:

a. ORP9 is usually recruited by PI4P binding to its N-terminal PH domain. Is there increased PI4P (GFP-P4Mx2) at the PM damage sites? If yes, the authors might want to take one step further to determine if such an increase is dependent on PI4K3A or PI4K2A?

Response: Thanks for this point. We have now examined PI4P, which is indeed enriched at damage sites (new Figure 3I-3J). Although not required by the editors, we also examined the potential kinases. We observed enrichment of PI4K2a, but not PI4K2b at damage sites (new Figure 3K-L). However, knocking down PI4K2a did not impact PI4P enrichment and the recruitment of ORP9 (Fig. S3F-3G). We suspect other kinases may be involved. This will be investigated further in the future.

b. Is calcium leakage into the pore required for the recruitment of ORP9 (and potentially PI4P)?

Response: thanks for the point. As suggested by the editors, this will be investigated in future studies.

c. Surprisingly, the authors showed that ORP9S which does not carry a PH domain is recruited to damaged PM more strongly than ORP9L. What are the explanations?

Response: With its PH domain and FFAT motif, ORP9L is more confined to membrane contact sites. ORP9S, without the PH domain, is more mobile and may serve as a “emergency response unit” that is better suited for quick response to repair membrane damages. Please note that

ORP9S is very similar to ORP2, which has 4 predicted positively charged patches that mediate PM targeting (see PMID: 30581148, Fig. 6 and 7). Thus, ORP9S can also be recruited to the damage site through PI4P, and through other negatively charged lipids such as PS.

d. The data suggest that ORP9 is recruited indirectly through other proteins, e.g., ORP10/11 which are known to dimerize with ORP9. Are ORP10/11 recruited to the damage sites? Does ORP9 recruitment require ORP10 or ORP11?

Response: We respectfully disagree that ORP9 is recruited indirectly through other proteins. As discussed above, both ORP9L and ORP9S can be recruited to the PM by interacting with negatively charged membrane lipids. Nevertheless, we examined the recruitment of ORP10 and ORP11, and their role in ORP9 function as suggested. Both ORP10 and ORP11 are efficiently recruited to the damage sites, but they are not required for ORP9 recruitment (see below). We are now carrying out many additional experiments to understand the roles of ORP10/11 in PM repair. We hope to exclude the data below from the current manuscript as these data do not improve its core discoveries. We seek your understanding on this.

2. The localization of ORP9 to the plasma membrane is minimal if any under basal conditions (Fig. 3C-F). The observed subtle increase in the TIRF signal appears to come from higher expression levels. These data do not support the main conclusions of this paper and are suggested to be removed.

Response: We would like to point out that data contained in Fig. 3C-3D are not derived from overexpressed ORP9. The association of ORP9 with the PM does occur at steady state as revealed by the functional changes (Figure 1-2). Furthermore, all quantifications of TIRF intensity were normalised to whole cell fluorescence intensity, to control for variation in expression. We hope to keep these data in supplemental figures. We sincerely seek your understanding on this.

3. Regarding ORP9-mediated PM repair:

a. Does the repair require the PS transfer activity of ORP9? This can be examined by rescuing experiments using the PS transfer-defective mutants.

Response: Thanks for the suggestion. We have now generated new data to show that the PS transfer defective mutants cannot efficiently restore PS enrichment at damage sites (new figure 5H-5J).

b. Based on the data presented, it is unclear how ORP9-mediated PS transfer promotes PM repair. If the authors propose tetraspanins as PS effectors, more insights into PS regulation of tetraspanins would be helpful in establishing the connection. Is the recruitment of other tetraspanins such as CD63 and CD151 regulated by ORP9?

Response: Great suggestions but this will be investigated in the future as suggested by the editors.

c. In the discussion, the authors suggested that due to the high basal levels of PS on the plasma membrane, PS should not be a messenger for PM repair. However, they also argued that PS might be regulating TSPAN4 enrichment at the damage sites. The fact that the PS probe is substantially increased at the damage sites in a ORP9-dependent manner suggests that a regulatory function of PS in PM repair requires much higher local concentrations of PS. It seems likely that similar to lysosomal repair in the PITT pathway, ATG2 might also be activated here by increased local PS levels and the high membrane curvature at the damage pore. The authors can quickly assess EGFP-ATG2 recruitment to the damage sites by live cell imaging in wild type and ORP9-KO cells.

Response: Thanks, and yes, we have carried out additional experiments on ATG2A (new Figure 5F-5G). ATG2A was indeed efficiently recruited to the damage sites, but its recruitment is extremely fast and does not require ORP5, ORP9 and PSS1. ATG2 may indeed play an important role in PM repair but this will be examined in future studies.

4. Is ORP9 recruited to the membrane pore upon PM damage by detergent or pyroptosis?

Functionally, does ORP9-KO render cells vulnerable to detergent treatment or pyroptosis induction?

Response: great suggestion. We have now found that ORP9 KO cells are more vulnerable to detergent treatment (Figure 4C)

5. Although not a focus in the current version of this study, the authors might want to discuss potential cholesterol transfer to the PM damage sites by OSBP. The specific enrichment of tetraspanins at the damage ring (TEMAs) suggests much higher local cholesterol levels than undamaged PM.

Response: Thanks. Cholesterol is known to impact the assembly of tetraspanins. Our work here focuses on phosphatidylserine, a previously unknown and therefore novel factor for tetraspanin function. It should also be noted that the level of plasma membrane cholesterol increased in ORP9 KO cells as we showed in the original Figure S2. Thus, the disruption of TEMA assembly in ORP9 KO cells is very unlikely caused by cholesterol deficiency. Our data suggest an independent role for PS in Tspan4 assembly. As suggested by the editors, the role of cholesterol during PM repair will be carefully examined in follow-up studies.

Minor:

6. Fig. 2B/C/E need to be supported by flow cytometry.

Response: We thank the reviewer for the suggestion. This staining method for PM phospholipids has been developed for fixed cells attached to a coverslip and may not be amenable to cells in suspension. To complement these data, we also utilised orthogonal methods including GFP-P4Mx2 coupled with TIRF imaging, and GFP-KRAS_{G12V} and mCherry-CAAX coupled with confocal microscopy to confirm the findings. Thus, given the concordance of the findings from two distinct approaches for both plasma membrane PI4P and PS in ORP9KO cells, we have not pursued the optimisation of the stains for flow cytometry at this point.

7. Fig. 2D needs quantification.

Response: thanks. We have since added a quantification for this figure.

Reviewer #3

Yang E. Li, Dougall Norris and their their colleagues reported a study named "ORP9 regulates plasma membrane repair by controlling phosphatidylserine and Tetraspanins ". It is indeed interesting to learn about their findings regarding ORP9 as a new regulator of phosphatidylserine levels on the plasma membrane and its role in efficient plasma membrane

repair. The presence of Tetraspanin-enriched rings and the requirement of functional ORP9 for the formation of both phosphatidylserine- and Tetraspanin-enriched rings highlight a novel mechanism for plasma membrane repair. However, several issues need to be addressed before publishing the study in the Journal of Cell Biology. Here are some possible insights:

Response: thanks for these positive and insightful comments.

Major points:

1. Regarding the origin of phosphatidylserine (PS) and its transfer to the plasma membrane in the context of membrane repair, the study suggests that ORP9 may play a role in delivering PS to the damaged site on the plasma membrane. While the exact internal membrane sources of PS during membrane repair are not specified in the information you provided, it is plausible that PS could be sourced from the endolysosomes, ER, or potentially other internal membrane compartments. In the figure 4K and 4L, the disruption of Golgi apparatus structure by BFA almost did not have impact on the ring structure, do you think PS from endolysosomes or ER would participant in the membrane repair process?

Response: We think it is most likely that the PS is newly synthesized from the ER. Both ORP9L and ORP9S have the FFAT motif, which targets proteins to the ER. Therefore, ORP9L and 9S should very likely function at ER-PM contact sites during PM damage to deliver PS from ER to the PM damage sites. Under normal conditions, most of ORP9L localizes to Golgi through the interaction between its PH domain and PI4P, which is enriched on the Golgi. A small portion of ORP9L can localize to the endolysosomes, but ORP9L is known to deliver PS to the endolysosomes but not to extract PS from there.

We have now also examined TEMA ring formation at damage sites in cells lacking the major PS synthase, PSS1, which localizes to the ER. In PSS1 KO cells, the formation of TEMA ring is significantly reduced (new Figure 5D), further supporting the core conclusion of this study that phosphatidylserine contributes to PM repair in part through TEMA assembly.

2. ORP9 involved PS transferring to the plasma membrane seems happen under normal state, so how about when the plasma membrane was locally damaged? The PS enriched in the ring structure is locally transferred from internal membrane parts at the precise damage site or laterally recruited from the other parts of plasma membrane.

Response: As mentioned above, most likely the PS is derived from the ER. ORP proteins are not known to transfer proteins from within the same membrane. In fact, the transfer of PS to the PM by ORPs requires the counter transfer of PI4P from PM to the ER, where PI4P is hydrolyzed.

Therefore, it is highly unlikely that ORP9 extracts PS from other parts of the PM to concentrate it at PM damage sites.

3.As the authors have mentioned that ORP5 is well known to localize to ER-PM contact sites to deliver phosphatidylserine from ER to the PM in exchange for PI4P and PI(4,5)P2. But, ORP5 didn't participant in the membrane damage repair. So i'm wondering that ORP9 has a specific involvement in the repair mechanism, possibly beyond just the transfer of PS, such as recruiting Tspan proteins and lipids?

Response: Thanks for this great point. We have now examined the role of ORP5 in TEMA formation (Fig. 5B). Surprisingly, while ORP5 is not recruited to damage sites, it is absolutely required for the formation of PS and TEMA rings. This data, together with data from PSS1 KO cells, do point to a crucial role for PS. We think that for PS ring formation, a basal level of PS at the PM is required, which is maintained by the constitutive activity of ORP5. During the damage, more PS is needed so ORP9 and possibly other PS transporters are then recruited.

4.Referring to the potential ability of ORP9 transferring PS to PM at the expense of PI4P, it is suggested to learn more about the PI4P distribution and concentration level under membrane damage repair process, especially at the ring structure.

Response: thanks, and yes, we have now examined PI4P, which is indeed enriched at damage sites (new Figure 3I-3J). Although not required by the editors, we also examined the potential kinases. We observed enrichment of PI4K2a, but not PI4K2b at damage sites (new Figure 3K-L). However, knocking down PI4K2a did not impact PI4P enrichment and the recruitment of ORP9 (Fig. S3F-3G). We suspect other kinases may be involved. This will be investigated further in the future.

January 2, 2024

Re: JCB manuscript #202307041R

Prof. Hongyuan Yang
UNSW Sydney
School of Biotechnology and Biomolecular Sciences, University of New South Wales
Sydney 2052
Australia

Dear Prof. Yang,

Thank you for submitting your revised manuscript entitled "Phosphatidylserine regulates plasma membrane repair through tetraspanin-enriched macrodomains". The manuscript has been seen by the original reviewers whose full comments are appended below. While the reviewers continue to be overall positive about the work in terms of its suitability for JCB, some important issues remain.

The original essential concern to examine ORP9 transfer activity (reviewer 2 point 3a) "Does the repair require the PS transfer activity of ORP9? This can be examined by rescuing experiments using the PS transfer-defective mutants" has not been completely addressed. We appreciate that you have added in new data showing that the PS transfer defective mutants cannot restore PS. However, we agree with the concern of reviewer 2 noted below that it is essential to demonstrate that the ORP9 transfer mutant cannot rescue PM repair in the knockout cells.

You will see that reviewer 1 has a number of remaining concerns. We appreciate the reviewer's feedback and agree that the proposed experiments would all strengthen your study. While we would welcome experimental data addressing these points, we recognize that there are also potential caveats to methods such as knock-sideways or induced degron approaches, therefore this is not required. As the request for double labeling and SIM/AiryScan was not made during the initial review, it is not expected at this point. Regarding the method of ORP9L and 9S localization, you have tested what had been editorially noted as essential in our initial decision letter, therefore further data is not required for this study. However, you must fully respond to all concerns raised by the reviewer, including toning down any conclusions and discussing alternative mechanisms, as well as the issue with inconsistencies in images.

Our general policy is that papers are considered through only one revision cycle; however, given that the suggested changes are relatively minor we are open to one additional short round of revision to fully address these critical issues. Please note that I will expect to make a final decision without additional reviewer input upon resubmission.

Please submit the final revision within one month, along with a cover letter that includes a point by point response to the remaining reviewer comments.

Thank you for this interesting contribution to Journal of Cell Biology. You can contact me or the scientific editor listed below at the journal office with any questions at cellbio@rockefeller.edu.

Sincerely,

Li Yu, PhD
Monitoring Editor

Andrea L. Marat, PhD
Senior Scientific Editor

Journal of Cell Biology

Reviewer #1 (Comments to the Authors (Required)):

This is a potentially important and exciting story. Unfortunately, a lack of mechanism, key control experiments, and appreciation for subtle differences in the probes preclude my support for its publication. The data presented do not support the title and narrative of the study. None of the data shows a direct role of PS regulating Tspan4 oligomerization / TEMA formation.

Conceptually, two items must be considered.

First, the impact of chronic depletion of ORP9 and ORP5 on the plasma membrane lipidome and abundance of Tspan4. The ORP9-KO cells have changes in the plasma membrane lipids: decreased PS, increase in PI4P, increased cholesterol in the outer leaflet, and likely a host of other changes that should be documented. It is not considered that these changes could influence the abundance of Tspan4 in the plasma membrane, its lateral mobility, and its ability to oligomerize. Any of these could limit the formation of the TEMA structures, and as there is no direct evidence of how PS influences Tspan4, these are essential controls. This would also explain the results with the ORP5-KO, which is equally vital for Tspan4 recruitment even though ORP5 is not actively recruited to the damage site. It remains to be seen if ORP9L and 9S are critical in the actual repair process. As mentioned in the previous review, the only way to figure this out is to use a knock-sideways or inducible degron approach. These approaches are commonplace in high-profile cell biology publications.

Second, there is a clear difference between the Tspan4 ring and the space occupied within it; the authors need to appreciate this observation. These are not diffraction-limited structures, and the images are apparent. The Tspan4, CHMP4B, and OSBP-PH (PI4P) all clearly depict a ringed structure. Yet, the LactC2, ATG2, ORP9L and S constructs are "filled circles" likely within the ring. In fact, some images (Fig 3B for instance) even show a void in GFP-ORP9 (reduced GFP signal) where the Tspan4 ring may reside. Double label should be done, and SIM/AiryScan or similar would be very beneficial. From these results, an alternative explanation is that PI4P and Tspan4 form a ringed structure around the liquid-ordered damaged zone to prevent its spread. The role of PS accumulation within the ring's interior would likely have alternative functions. The more important function of ORP9 may be to remove PI4P from the ring's exterior to preserve a gradient to support the damage-limiting Tspan4 ring.

The discussion needs insight and an appreciation for the biophysical properties of lipids and membranes. The ORP5-KO does not accumulate at the sites of damage, yet ORP5-KO cells have a reduction in PS at the damage site. Thus, an alternative mechanism would be that plasmalemmal PS laterally diffuses into that damage site and remain there due to a combination of retention or increased dwell time in the liquid-ordered damage site and the Tspan4 acting as a diffusional barrier.

If ORP9L and 9S target the sites by binding PI4P, they should look like OSBP; they do not. The authors need to demonstrate how 9L and 9S localize to the site. I agree with the other reviewers, it may require binding to another plasma membrane/damage site protein.

Technical issues:

The epi images in Figures 1 and 2 are saturated and vastly over-enhanced. If these images were used for the quantitation, the results would be meaningless.

Figure 4C was included in response to another reviewer. However, while this is statistically significant due to a large sample number, it's difficult to imagine it is biologically significant.

Inconsistency of the images. Some look like confocal slices through the cell, others look like confocal slices near the base of the cell, and others are TIRF. 5A and 5B (Tspan4) look nothing a like, 4F and 5A (LactC2) also look different.

Reviewer #2 (Comments to the Authors (Required)):

The new data in the revised manuscript have strongly strengthened the conclusions. I don't have any major concerns. I recommend publication of this paper upon the completion of one more critical experiment to confirm that the PS transfer-defective mutants of ORP9 cannot rescue PM repair in the knockout cells.

Reviewer #3 (Comments to the Authors (Required)):

The authors have successfully addressed the point I raised, so I recommend the publication of this intriguing revised manuscript in the Journal of Cell Biology after making slight modifications as outlined below.

Minor points:

In Figure 4H and Figure S3K, the exact time information should be added.

Point-by-point rebuttal

Editorial comments:

The original essential concern to examine ORP9 transfer activity (reviewer 2 point 3a) "Does the repair require the PS transfer activity of ORP9? This can be examined by rescuing experiments using the PS transfer-defective mutants" has not been completely addressed. We appreciate that you have added in new data showing that the PS transfer defective mutants cannot restore PS. However, we agree with the concern of reviewer 2 noted below that it is essential to demonstrate that the ORP9 transfer mutant cannot rescue PM repair in the knockout cells.

Response: Thanks. We have now added the new data as requested (New figure 4A-4C). The mutants showed much weaker rescue effects than WT proteins, further suggesting the PS binding/transfer activity of ORP9 is critical for efficient PM repair. Please note that the L/D mutants may still retain some PS transfer activity and the mutants also showed higher enrichment at damaged sites (Fig. 3D-3F). Thus, the L/D mutants could still transfer a small amount of PS to the damaged sites. While this limited amount of PS may fall below the detection limit of Lact-C2, it could help slow down the entry of the propidium iodide.

You will see that reviewer 1 has a number of remaining concerns. We appreciate the reviewer's feedback and agree that the proposed experiments would all strengthen your study. While we would welcome experimental data addressing these points, we recognize that there are also potential caveats to methods such as knock-sideways or induced degron approaches, therefore this is not required. As the request for double labeling and SIM/AiryScan was not made during the initial review, it is not expected at this point. Regarding the method of ORP9L and 9S localization, you have tested what had been editorially noted as essential in our initial decision letter, therefore further data is not required for this study. However, you must fully respond to all concerns raised by the reviewer, including toning down any conclusions and discussing alternative mechanisms, as well as the issue with inconsistencies in images.

Response: Thanks. We have addressed each point below, carried out new experiments and made relevant changes in the discussion.

Reviewer #1 (Comments to the Authors (Required)):

This is a potentially important and exciting story. Unfortunately, a lack of mechanism, key control experiments, and appreciation for subtle differences in the probes preclude my support for its publication. The data presented do not support the title and narrative of the study. None of the data shows a direct role of PS regulating Tspan4 oligomerization / TEMA formation.

First, the impact of chronic depletion of ORP9 and ORP5 on the plasma membrane lipidome and abundance of Tspan4. The ORP9-KO cells have changes in the plasma membrane lipids: decreased PS, increase in PI4P, increased cholesterol in the outer leaflet, and likely a host of

other changes that should be documented. It is not considered that these changes could influence the abundance of Tspan4 in the plasma membrane, its lateral mobility, and its ability to oligomerize. Any of these could limit the formation of the TEMA structures, and as there is no direct evidence of how PS influences Tspan4, these are essential controls. This would also explain the results with the ORP5-KO, which is equally vital for Tspan4 recruitment even though ORP5 is not actively recruited to the damage site. It remains to be seen if ORP9L and 9S are critical in the actual repair process. As mentioned in the previous review, the only way to figure this out is to use a knock-sideways or inducible degron approach. These approaches are commonplace in high-profile cell biology publications.

Response: We acknowledge the possibility that additional lipids may play a role in the assembly of the TEMA ring. However, it is impossible to examine the possible changes of all PM lipids and their roles in TEMA assembly and PM repair in ORP5/9 KO cells. While we recognize that the most direct biochemical evidence for PS in TEMA formation is lacking (now added to the discussion), our data strongly support a role for PS in forming the TEMA ring. We wish to highlight a few points that may have been missed by this reviewer. 1) since the **key shared function of ORP5 and ORP9** is to deliver PS to the PM, the requirement of both ORP5 and ORP9 for TEMA function further points to a critical role for PS, but not other lipids in TEMA function. 2) **reduced PS synthesis also impaired TEMA assembly** (Fig. 5). This is **also a key piece of evidence** supporting PS but not other lipids in TEMA assembly. 3) plasma membrane cholesterol, a well-established lipid to facilitate the function of tetraspanins, was increased (not decreased) in ORP9 KO cells (Fig. S2) and in PSS1-deficient cells (Li et al., 2021; Trinh et al., 2020; Trinh et al., 2022). Therefore, the impaired TEMA formation is unlikely to be related to cholesterol.

Second, there is a clear difference between the Tspan4 ring and the space occupied within it; the authors need to appreciate this observation. These are not diffraction-limited structures, and the images are apparent. The Tspan4, CHMP4B, and OSBP-PH (PI4P) all clearly depict a ringed structure. Yet, the LactC2, ATG2, ORP9L and S constructs are "filled circles" likely within the ring. In fact, some images (Fig 3B for instance) even show a void in GFP-ORP9 (reduced GFP signal) where the Tspan4 ring may reside. Double label should be done, and SIM/AiryScan or similar would be very beneficial. From these results, an alternative explanation is that PI4P and Tspan4 form a ringed structure around the liquid-ordered damaged zone to prevent its spread. The role of PS accumulation within the ring's interior would likely have alternative functions. The more important function of ORP9 may be to remove PI4P from the ring's exterior to preserve a gradient to support the damage-limiting Tspan4 ring.

Response: As illustrated in Figure 3I, Figure 4G, and 5A, **both PI4P and PS exhibit a 'filled circle' and a ring at the early stages of observation**. However, the central region of the PI4P-accumulating patch diminishes over time, contrasting with the sustained accumulation of both the central region and the ring in the case of PS during the repair phase. This dynamic alteration aligns with the functional role of ORP9 at the plasma membrane, as elucidated in Figure 1 and 2. It is possible that ORP9 delivers the PS to this central zone from which it diffuses into the surrounding ring-like structure. This will be a focus of future studies. Finally,

the well-established functions of PI4P are to recruit proteins and to serve as a driver for ORPs function. Although a physical role for PI4P in restricting the lateral diffusion of lipids as the reviewer suggested cannot be completely ruled out, it remains a rather remote possibility.

The discussion needs insight and an appreciation for the biophysical properties of lipids and membranes. The ORP5-KO does not accumulate at the sites of damage, yet ORP5-KO cells have a reduction in PS at the damage site. Thus, an alternative mechanism would be that plasmalemmal PS laterally diffuses into that damage site and remain there due to a combination of retention or increased dwell time in the liquid-ordered damage site and the Tspan4 acting as a diffusional barrier.

Response: Again, given the key shared function between ORP5 and ORP9 in PS transport, we propose that PS delivery constitutes the most likely repair mechanism by these proteins. Moreover, reduced PS synthesis also impaired TEMA assembly (Fig. 5). A role for TEMA ring to keep PS within the damaged sites is very much out of the scope of our study and is not supported by our data at all.

If ORP9L and 9S target the sites by binding PI4P, they should look like OSBP; they do not. The authors need to demonstrate how 9L and 9S localize to the site. I agree with the other reviewers, it may require binding to another plasma membrane/damage site protein.

Response: A key discovery of our current work is the plasma membrane localization of ORP9, but not OSBP (Fig. 1 and S1). Therefore, **a difference in distribution pattern between ORP9 and OSBP-PH is expected**. Other proteins or lipids, e.g. PIP2 should also help recruit ORP9, but not OSBP-PH to the PM. These will be investigated in future studies.

Technical issues:

The epi images in Figures 1 and 2 are saturated and vastly over-enhanced. If these images were used for the quantitation, the results would be meaningless.

Response: We appreciate the reviewer's concern. It is important to note that the images utilized for quantification purposes are not saturated. The change in the enrichment of P4Mx2 at intercellular sites with ORP9 overexpression is so great that in order to show the localization under control conditions, we had to over-saturate the images. We only captured images of cells within a similar range of whole-cell brightness and disregarded any cells with over-saturated areas.

Figure 4C was included in response to another reviewer. However, while this is statistically significant due to a large sample number, it's difficult to imagine it is biologically significant.

Response: We have confidence in the reliability of the paired t-test to yield statistically meaningful results.

Inconsistency of the images. Some look like confocal slices through the cell, others look like confocal slices near the base of the cell, and others are TIRF. 5A and 5B (Tspan4) look nothing a like, 4F and 5A (LactC2) also look different.

Response: All imaging experiments of plasma membrane damage were taken on the same confocal microscope in the bottom plane of the cell, and thus these images were not TIRF images. The previous figure 5A was taken at a higher resolution using Airyscan with a smaller damage region, hence the discrepancy between the morphology with the other experiments listed. **We have since re-performed this experiment** with a larger damage size, consistent with previous images and have replaced the selected representative images for Figure 5A.

Reviewer #2 (Comments to the Authors (Required)):

The new data in the revised manuscript have strongly strengthened the conclusions. I don't have any major concerns. I recommend publication of this paper upon the completion of one more critical experiment to confirm that the PS transfer-defective mutants of ORP9 cannot rescue PM repair in the knockout cells.

Response: Thanks. Please see response as above.

Reviewer #3 (Comments to the Authors (Required)):

The authors have successfully addressed the point I raised, so I recommend the publication of this intriguing revised manuscript in the Journal of Cell Biology after making slight modifications as outlined below.

Minor points:

In Figure 4H and Figure S3K, the exact time information should be added.

Response: Thanks. We have mentioned the time point (10 minutes) in the figure legend.

February 24, 2024

RE: JCB Manuscript #202307041RR

Prof. Hongyuan Yang
UNSW Sydney
School of Biotechnology and Biomolecular Sciences, University of New South Wales
Sydney 2052
Australia

Dear Prof. Yang:

Thank you for addressing the final reviewer points for your manuscript "Phosphatidylserine regulates plasma membrane repair through tetraspanin-enriched macrodomains". We would be happy to publish your paper in JCB pending final revisions necessary to meet our formatting guidelines (see details below). In your final revision, please discuss the alternative interpretations that the KO phenotype could be explained by a change in plasma membrane composition, or PS could be enriched at the damage site by another mechanism.

A. MANUSCRIPT ORGANIZATION AND FORMATTING:

- 1) Text limits: Character count for Reports is < 20,000, not including spaces. Count includes abstract, introduction, *combined results and discussion*, and acknowledgments. Count does not include title page, figure legends, materials and methods, references, tables, or supplemental legends.
- 2) Figures limits: Reports may have up to 5 main text figures.
- 3) Figure formatting: Scale bars must be present on all microscopy images, including inset magnifications. Molecular weight or nucleic acid size markers must be included on all gel electrophoresis. * In order to accommodate readers with red-green color blindness, we ask that you please avoid the use of red/green color schemes.
- 4) Statistical analysis: Error bars on graphic representations of numerical data must be clearly described in the figure legend. The number of independent data points (n) represented in a graph must be indicated in the legend. Statistical methods should be explained in full in the materials and methods. For figures presenting pooled data the statistical measure should be defined in the figure legends. Please also be sure to indicate the statistical tests used in each of your experiments (either in the figure legend itself or in a separate methods section) as well as the parameters of the test (for example, if you ran a t-test, please indicate if it was one- or two-sided, etc.). Also, if you used parametric tests, please indicate if the data distribution was tested for normality (and if so, how). If not, you must state something to the effect that "Data distribution was assumed to be normal but this was not formally tested."
- 5) Abstract and title: The abstract should be no longer than 160 words and should communicate the significance of the paper for a general audience. The title should be less than 100 characters including spaces. Make the title concise but accessible to a general readership.
- 6) Materials and methods: Should be comprehensive and not simply reference a previous publication for details on how an experiment was performed. Please provide full descriptions in the text for readers who may not have access to referenced manuscripts. * For example, please describe the PM PI4P/PS labeling protocol.
- 7) * All antibodies, cell lines, animals, and tools used in the manuscript should be described in full, including accession numbers for materials available in a public repository such as the Resource Identification Portal. Please be sure to provide the sequences for all of your primers/oligos and RNAi constructs in the materials and methods. You must also indicate in the methods the source, species, and catalog numbers (where appropriate) for all of your antibodies. Please also indicate the acquisition and quantification methods for immunoblotting/western blots. *
- 8) Microscope image acquisition: The following information must be provided about the acquisition and processing of images:
 - a. Make and model of microscope
 - b. Type, magnification, and numerical aperture of the objective lenses

- c. Temperature
- d. Imaging medium
- e. Fluorochromes
- f. Camera make and model
- g. Acquisition software
- h. Any software used for image processing subsequent to data acquisition. Please include details and types of operations involved (e.g., type of deconvolution, 3D reconstitutions, surface or volume rendering, gamma adjustments, etc.).

10) Supplemental materials: There are strict limits on the allowable amount of supplemental data. Reports may have up to 3 supplemental figures. Please also note that tables, like figures, should be provided as individual, editable files. A summary of all supplemental material should appear at the end of the Materials and methods section.

13) ORCID IDs: ORCID IDs are unique identifiers allowing researchers to create a record of their various scholarly contributions in a single place. Please note that ORCID IDs are now *required* for all authors. At resubmission of your final files, please be sure to provide your ORCID ID and those of all co-authors.

Please note that JCB now requires authors to submit Source Data used to generate figures containing gels and Western blots with all revised manuscripts. This Source Data consists of fully uncropped and unprocessed images for each gel/blot displayed in the main and supplemental figures. Since your paper includes cropped gel and/or blot images, please be sure to provide one Source Data file for each figure that contains gels and/or blots along with your revised manuscript files. File names for Source Data figures should be alphanumeric without any spaces or special characters (i.e., SourceDataF#, where F# refers to the associated main figure number or SourceDataFS# for those associated with Supplementary figures). The lanes of the gels/blots should be labeled as they are in the associated figure, the place where cropping was applied should be marked (with a box), and molecular weight/size standards should be labeled wherever possible.

Journal of Cell Biology now requires a data availability statement for all research article submissions. These statements will be published in the article directly above the Acknowledgments. The statement should address all data underlying the research presented in the manuscript. Please visit the JCB instructions for authors for guidelines and examples of statements at (<https://rupress.org/jcb/pages/editorial-policies#data-availability-statement>).

B. FINAL FILES:

****It is JCB policy that if requested, original data images must be made available to the editors. Failure to provide original images upon request will result in unavoidable delays in publication. Please ensure that you have access to all original data images prior to final submission.****

****The license to publish form must be signed before your manuscript can be sent to production. A link to the electronic license to publish form will be sent to the corresponding author only. Please take a moment to check your funder requirements before choosing the appropriate license.****

Thank you for this interesting contribution, we look forward to publishing your paper in Journal of Cell Biology.

Sincerely,

Li Yu, PhD
Monitoring Editor

Andrea L. Marat, PhD
Senior Scientific Editor

Journal of Cell Biology